# Uncertainty of Thoughts: Uncertainty-Aware Planning Enhances Information Seeking in LLMs

**Zhiyuan Hu**[1]* **Chumin Liu**[2] **Xidong Feng**[3] **Yilun Zhao**[4]
**See-Kiong Ng**[1] **Anh Tuan Luu**[2] **Junxian He**[5] **Pang Wei Koh**[6] **Bryan Hooi**[1]

[1] National University of Singapore    [2] Nanyang Technological University
[3] University College London    [4] Yale University
[5] The Hong Kong University of Science and Technology    [6] University of Washington

## Abstract

In the face of uncertainty, the ability to *seek information* is of fundamental importance. In many practical applications, such as medical diagnosis and troubleshooting, the information needed to solve the task is not initially given, and has to be actively sought by asking follow-up questions (for example, a doctor asking a patient for more details about their symptoms). In this work, we introduce Uncertainty of Thoughts (UoT), an algorithm to augment large language models with the ability to actively seek information by asking effective questions. UoT combines 1) an *uncertainty-aware simulation approach* which enables the model to simulate possible future scenarios and how likely they are to occur, 2) *uncertainty-based rewards* motivated by information gain which incentivizes the model to seek information, and 3) a *reward propagation scheme* to select the optimal question to ask in a way that maximizes the expected reward. In experiments on medical diagnosis, troubleshooting and the '20 Questions' game, UoT achieves an average performance improvement of 38.1% in the rate of successful task completion across multiple LLMs compared with direct prompting, and also improves efficiency (i.e., the number of questions needed to complete the task). Our code are released[2].

## 1 Introduction

As the capabilities of large language models (LLMs) grow, they are being increasingly deployed in challenging real-world settings involving uncertainty and ambiguity. In particular, recent work aims to develop LLM agents or assistants [36, 26] that effectively complete tasks in interactive environments, leading to a growing need for LLMs that can *actively seek the information they need* to solve a task by asking questions in conversational settings. For example, in medical diagnosis, patients often do not initially report their symptoms in full detail. In such situations, a doctor's ability to ask effective questions is crucial, as a successful diagnosis often depends on revealing important details that the patient did not initially provide (Figure 1).

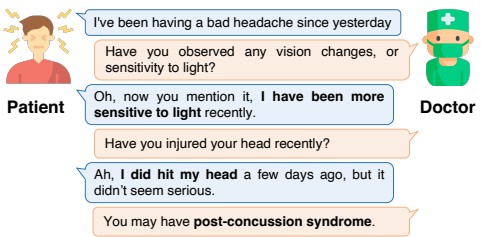

Figure 1: The importance of information seeking in medical diagnosis. The patient initially only complains of a headache, but by asking the right questions, the doctor uncovers the critical information needed for a correct diagnosis.

---

*Corresponding to: Zhiyuan Hu. Email: zhiyuan_hu@u.nus.edu
[2]https://github.com/zhiyuanhubj/UoT

Recent techniques aim to improve LLMs' reasoning or planning abilities based on the given information rather than enabling LLMs to seek information efficiently. For example, Chain-of-Thought (CoT) [35] and Tree-of-Thoughts (ToT) [38] allow LLMs to express intermediate 'thoughts' and reason over them. Unlike these methods, our focus is on enabling the LLM to ask questions effectively by explicitly guiding the model toward reducing uncertainty, which these do not consider. Thus, they lack effective signals for questions that better reduce uncertainty by revealing critical information.

To enhance LLMs in actively seeking information, we introduce Uncertainty of Thoughts (UoT), a plug-and-play approach that improves LLMs' abilities to ask useful questions by modeling their own uncertainty. UoT is a principled approach relying on *uncertainty-based rewards motivated by information gain*, which incentivizes a model to seek information in a way that maximally reduces the amount of information it does not know. To utilize these rewards, we develop an *uncertainty-aware simulation framework*, enabling the model to simulate possible future scenarios along with how likely they are to occur. Given these scenarios, we utilize a *reward propagation scheme* to select the optimal question to ask in a way that maximizes the expected reward.

Additionally, most standard benchmarks for LLMs, particularly in question answering, assume that all necessary information to solve a task is provided at the outset, and thus do not evaluate the model's active information-seeking capabilities. To close this gap, we first introduce a benchmark comprising 5 datasets[3] on 3 tasks: 20 Questions, a simplified medical diagnosis task, and a basic troubleshooting task. These tasks are designed to measure the model's ability to ask questions effectively to gather the information they need. For example, the 20 Questions game, also studied by Noever et al.[22], requires the model to ask 'yes' or 'no' questions to determine an unknown object or entity. This scenario serves as a clear and easily analyzed test case, isolating the model's ability to recognize its own uncertainty, and to ask questions that guide it to the correct answer.

Our work is a step toward LLMs that can effectively operate in settings with high uncertainty and ambiguity, beyond conventional QA settings where all the information needed to solve the task is provided to the model at the outset. To the best of our knowledge, UoT is the first approach for enabling LLMs to ask effective questions by explicitly modeling and seeking to reduce their uncertainty. Our key contributions are as follows:

1. We introduce Uncertainty of Thoughts (UoT), a plug-and-play approach enabling LLMs to explicitly model and seek to reduce their uncertainty. UoT utilizes a principled approach based on an uncertainty-aware framework for simulating possible futures, rewards motivated by information gain, and a reward propagation scheme to select the optimal question to ask.

2. We introduce a benchmark of 3 tasks and 5 datasets, designed to evaluate the ability of LLMs to seek the information they need by asking questions.

3. Experiments show that UoT improves the success rate of multiple LLMs by 38.1% on average compared with direct prompting, achieving top performance on both task success and efficiency. Our benchmark and code are publicly available.

## 2 Methodology

### 2.1 Problem Formulation

The problem setting involves two roles: the Questioner and the Answerer, performed by the LLM and a human, respectively. The goal of the Questioner is to deduce an unknown piece of information. We formulate this using a *possibility space* $\Omega$, which is the set of all possible options, of which a single element $\omega \in \Omega$, is the *true option* in each given scenario[4]. For example, in a medical diagnosis setting, $\Omega$ is the set of all possible diseases relevant in the context, e.g., $\Omega = \{\text{Bronchitis}, \text{Flu}, \ldots, \text{Hypertension}\}$, and for each patient, $\omega$ is the actual disease of the patient.

---

[3]We also incorporate the efforts of prior datasets [31, 37, 19, 27], through further work and refinement to construct this benchmark. Details are introduced in section 3 Experiments and Appendix I.2.

[4]Under the measure-theoretic formulation of probability, the *sample point* $\omega$ is an element of the *sample space* $\Omega$, and all random variables are defined to be functions of $\omega$. While we conform to this formulation, we try to avoid unnecessary measure-theoretic background for ease of understanding; hence, it is sufficient for readers to understand $\omega$ as the 'true option' in each scenario.

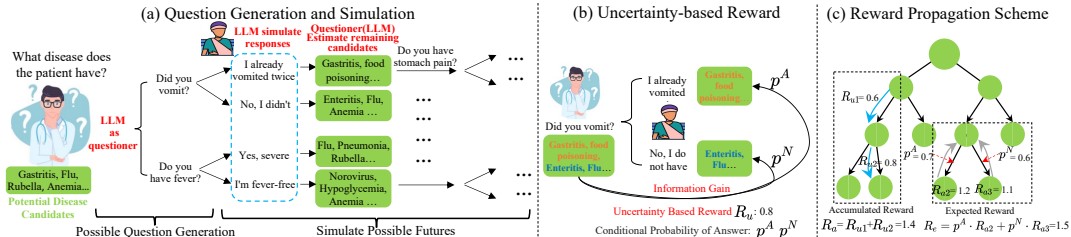

Figure 2: UoT Overview: UoT includes three components: (a) Question Generation and Simulation, where an LLM proposes candidate questions and simulates future scenarios; (b) Uncertainty-based Rewards, measuring the uncertainty reduction from answers to a question, and (c) Reward Propagation computing accumulated rewards $R_a$ over past questions, and expected rewards $R_e$ capturing expected future gains. The process ends by choosing questions with the highest expected reward.

The interaction between the Questioner and the Answerer occurs over multiple turns. For instance, the Questioner may ask, "Do you have a fever?", to which the Answerer responds, "Yes, I've had a high fever for the past two days." The Questioner then asks another question such as "Have you vomited?" This exchange continues either until the Questioner correctly determines the final answer, or the conversation reaches a maximum number of turns. At this point, the interaction ends, and the Questioner is *successful* if it has correctly determined the true option $\omega$.

Most of the description of our approach focuses on the *closed set* scenario, in which we assume that the Questioner starts with knowledge of the possibility space $\Omega$, e.g., the set of all possible diseases in medical diagnosis. In our extension section 2.7, we adapt our approach to the *open set* scenario, in which this knowledge is absent. Moreover, as the questioning progresses, we use an LLM to gradually refine this set of possibilities to those that are consistent with the current answers given so far by the Answerer. Define the *current possibility set* $\Omega_i$ as the subset of $\Omega$ that is consistent with all answers given by the Answerer before the start of the $i$th interaction step.

As we discuss more later, we focus on applications where answers can be grouped into a small number of semantically distinct categories (in our case, affirmative and negative responses), as this allows us to compute meaningful uncertainty metrics in a simpler way. Conceptually, our framework can straightforwardly be extended to allow for a wider selection of answers.

## 2.2 Uncertainty of Thoughts: Overview

As Figure 2 shows, to effectively reduce uncertainty, our UoT method first **generates multiple questions** as candidates to ask, and **simulates possible futures** for each one in the form of a tree structure. Next, **uncertainty-based rewards**, motivated by information gain, are used to assess the questions within the simulation. Finally, a **reward propagation scheme** is used to compute the expected reward from asking each candidate question, allowing us to select the one with highest expected reward, to ask the Answerer.

## 2.3 Question Generation and Simulation

UoT starts by using an LLM to generate several candidate questions, then simulates future scenarios for each one. This simulation process allows us to measure how much information we can expect to gain in the next few steps from each question, and thus to choose the most suitable question.

**Question Generation** Recall that our setting involves sequential interactions between a Questioner (e.g., a chatbot) and an Answerer (e.g., a human patient). During the $i$th interaction step, the Questioner generates candidate questions, then selects one of these to ask, denoted as $q_i$.

To generate candidate questions to ask, UoT uses two inputs: (1) the *history of past interactions* $h_i = \{q_1, a_1, q_2, a_2, \ldots, q_{i-1}, a_{i-1}\}$, comprising the sequence of past questions and answers; and (2) the *current possibility set* $\Omega_i$. These two inputs are combined to form a prompt that includes instructions explaining the nature of the task (e.g., how the 20 Questions game works), provides the current history $h_i$ and the current possibility set $\Omega_i$, and asks an LLM to generate $m$ candidate next questions, conditioned on the previous context. This prompt, denoted as $\mathsf{Prompt}_{\mathsf{gen}}(h_i, \Omega_i)$, is fed to

our generator $\mathsf{LLM}_{\text{gen}}$, which then generates $m$ candidate questions, denoted $q_i^1, q_i^2, \ldots, q_i^m$:

$$q_i^1, q_i^2, \ldots, q_i^m = \mathsf{LLM}_{\text{gen}}(\mathsf{Prompt}_{\text{gen}}(h_i, \Omega_i)) \tag{1}$$

**Multistep Simulation**    As shown in Figure 2 (a), the Question Generation stage generates candidate questions such as $q_i^1 = $ "Did you vomit?" Next, during Simulation stage, for each such generated candidate question, we simulate possible futures for a few steps, forming a tree of possibilities. This process enables us to compute rewards for each question, helping us to decide which question to ask.

Each node of the tree can be one of two types: *Answerer Nodes* where it is the Answerer's turn to answer a question, and *Questioner Nodes* where it is the Questioner's turn to ask a question. At the root, a question has just been asked (e.g., $q_i^1$), so the root is an Answerer Node. Next, we explain how to construct tree by recursively expanding (or 'branching') each node to construct its children, i.e., starting from the root, then proceeding to its children, and so on.

- At each **Answerer Node**, a question has just been asked. Next, we need to further 'branch' the tree based on the possible answers to the current question. Rather than allowing completely open-ended answers, we instead focus on affirmative and negative responses[5], as this allows us to compute meaningful uncertainty metrics, as we discuss later. Hence, we branch the node into two children, corresponding to affirmative and negative answers.
- At each **Questioner Node**, we prompt an LLM to generate $m$ questions using the current history and possibility set, in the same way as in the Question Generation step. Note that while the generation procedure is similar, the purpose is different: the Question Generation step generates *candidate* questions to select from, while here we are generating *simulated* questions to form a tree for the purpose of evaluating the current question. The resulting $m$ generated questions are added to the tree as children of the current node.

In this way, we recursively generate tree nodes, stopping at a fixed number of levels (i.e., depth).

While generating this tree, we also recursively compute the *current possibility set* $\Omega_v$ at each node $v$. Specifically, let $h_v$ be the current conversation history up to node $v$, combining both the actual conversation history $h_i$ and the simulated conversation up to node $v$. Then the current possibility set at this node, denoted $\Omega_v$, is the subset of the possibility space consistent with $h_v$. At the root, the current possibility set is only limited by the actual conversation history, i.e., $\Omega_i$. Then, as we proceed over the simulated tree, note that the current possibility set only changes at Answerer nodes, when an answer is added to the current history. Hence, at each Answerer node $v$, we prompt a new LLM (an 'Answerer Simulator' $\mathsf{LLM}_{\text{ans}}$), to determine the further subset $\Omega_v^A \subseteq \Omega_v$ for which the answer to the current question is affirmative, and the corresponding $\Omega_v^N = \Omega_v \setminus \Omega_v^A$ for which the answer is negative.[6] This allows us to recursively compute the possibility sets of the children of $v$ (which themselves correspond to the affirmative and negative answers).

$$\Omega_v^A, \Omega_v^N = \mathsf{LLM}_{\text{ans}}(\mathsf{Prompt}_{\text{ans}}(h_v, \Omega_v)) \tag{2}$$

In this way, we can recursively compute the possibility set on each node of the tree.

## 2.4    Uncertainty-Based Reward Calculation

To develop suitable information-seeking approaches, a critical question is *how to evaluate the effectiveness of a question, i.e., its contribution to reducing uncertainty*. To address this, we turn to information theory, specifically the concept of *information gain*, which measures the amount by which uncertainty decreases after a particular observation. To reward information-seeking behavior, we assign rewards to questions based on how much they reduce the model's uncertainty about the unknown random variable. These reward signals are used by our UoT framework to determine which question to select, to maximize the reduction of uncertainty.

**Entropy.** Entropy and information gain are well-known concepts in information theory [29]. In our work, we use these concepts to measure how much information is gained (or equivalently, how much

---

[5]As shown Figure 2 (a), for question 'Did you vomit?', possible affirmative responses include 'yes' or 'I already vomited twice', while negative responses could be 'no' or 'I don't have'.

[6]In practice, allowing overlap between $\Omega_v^A$ and $\Omega_v^N$ may be more realistic. However, in this work, we consider only the simplified scenario where they are disjoint.

uncertainty is reduced) by asking a question, to formulate our rewards. Entropy measures the level of uncertainty in a random variable: higher entropy indicates greater uncertainty. The entropy of a discrete random variable $X$ taking values $x_1, ..., x_n$ is:

$$H(X) = -\sum_{i=1}^{n} p(x_i) \log p(x_i) \tag{3}$$

Since our goal is to reduce the uncertainty in the unknown $\omega \in \Omega$, we use entropy to measure this uncertainty. Formally, let $\Omega = \{\omega_1, \cdots, \omega_n\}$, and we define an additional set of arbitrary real numbers $\mathcal{X} = \{x_1, \cdots, x_n\} \subseteq \mathbb{R}$ which we will associate with each of these possibilities. Define a random variable $X : \Omega \to \mathcal{X}$ such that $X(\omega_i) = x_i$. Intuitively, $X$ is a discrete random variable that takes the value $x_i$ if the $i$th possibility is true, i.e., if $\omega = \omega_i$. $X$ serves to capture our uncertainty about $\omega$, since observing $X$ is equivalent to observing the true option $\omega$. As a simple example, suppose our possibility space is $\Omega = \{\omega_1, \omega_2, \omega_3\}$; we accompany these with real numbers $x_1, x_2, x_3$, and have a distribution for our random variable $X$ reflecting prior beliefs over these possibilities: e.g., $p(x_1) = 0.2$, $p(x_2) = 0.3$, $p(x_3) = 0.5$. Conceptually, our framework allows for any prior probability distribution over the possibilities (i.e., $p(x_i)$), but in our experiments, we assume a uniform distribution over them due to the lack of an informative prior.

Before asking any questions, our uncertainty about the unknown $\omega$ is given by $H(X)$, as in Eq. (3). At any node $v$ of the trees described in the previous section, recall that we have a conversation history $h_v$ which contains some answers given by the Answerer. This history limits the current possibility set to those in $\Omega_v \subseteq \Omega$, thereby reducing our uncertainty. We model this using the standard notion of *conditional probability on an event*: since $\Omega_v \subseteq \Omega$, thus $\Omega_v$ is an event which we can condition on:

$$p(x_i|\Omega_v) = p(x_i)/p(\Omega_v) \quad \forall i \text{ such that } \omega_i \in \Omega_v \tag{4}$$

where $p(\Omega_v)$ is the sum of probabilities of the elements in $\Omega_v$. To illustrate, we continue from the earlier example, where $p(x_1) = 0.2$, $p(x_2) = 0.3$, $p(x_3) = 0.5$. If the conversation history $h_v$ at node $v$ is only consistent with $x_1$ and $x_2$, i.e., $\Omega_v = \{\omega_1, \omega_2\}$, we can adjust probability distribution by conditioning: e.g., the adjusted probability of $x_1$ is $p(x_1)/p(\Omega_v) = 0.2/(0.2 + 0.3) = 0.4$.

Next, to quantify the uncertainty at node $v$, note that since $X$ is conditionally distributed based on $p(\cdot|\Omega_v)$, the entropy of this distribution is:

$$H_v(X) := \sum_{i:\omega_i \in \Omega_v} p(x_i|\Omega_v) \log p(x_i|\Omega_v) \tag{5}$$

Intuitively, $H_v(X)$ is the remaining uncertainty in $X$ at node $v$ (i.e., after observing the history $h_v$).

**Information Gain at a Node** We now quantify the uncertainty reduction when receiving answers at an Answerer node $v$. Recall that the answer given at $v$ partitions $\Omega_v$ into two disjoint subsets: $\Omega_v = \Omega_v^A \cup \Omega_v^N$, where $\Omega_v^A$ and $\Omega_v^N$ are the subsets of possibilities resulting in affirmative and negative answers to last asked question. Given an affirmative answer, the remaining entropy becomes:

$$H_v^A(X) := \sum_{i:\omega_i \in \Omega_v^A} p(x_i|\Omega_v^A) \log p(x_i|\Omega_v^A) \tag{6}$$

We define $H_v^N(X)$ analogously for negative answers. Let $p_v^A = p(\Omega_v^A)/p(\Omega_v)$ and $p_v^N = p(\Omega_v^N)/p(\Omega_v)$ be the conditional probabilities of affirmative and negative answers at node $v$. To compute the expected entropy after receiving the answer at node $v$, since we have a $p_v^A$ probability of receiving an affirmative answer and $p_v^N$ of a negative answer, the expected entropy is:

$$p_v^A \cdot H_v^A(X) + p_v^N \cdot H_v^N(X) \tag{7}$$

As such, the expected information gain at node $v$ is the difference in entropies before and after receiving the answer:

$$IG_v(X) := H_v(X) - p_v^A \cdot H_v^A(X) - p_v^N \cdot H_v^N(X) \tag{8}$$

We can simplify this: as proven in Appendix A, the above equation reduces to:

$$IG_v(X) = -p_v^A \log p_v^A - p_v^N \log p_v^N \tag{9}$$

This represents the expected reduction of uncertainty in $X$ when receiving an answer at node $v$. Note that it has an entropy-like expression, and is therefore nonnegative.

**Reward Formulation**    A natural approach would be to define the reward function $R_u(v)$ at node $v$ as the information gain $IG_v(X)$: that is, the reward from the question at node $v$ is the expected information gain $IG_v(X)$ from receiving its answer. In practice, we find that a slightly modified function $\widetilde{IG}_v(X)$ is preferable. In particular, we find that $IG_v(X)$ does not result in sufficiently sharp differences in reward over the typical ranges we encounter. Hence, we introduce an additional hyperparameter $\lambda \geq 0$ which helps to sharpen the rewards using a scaling approach. We compare other scaling methods and determine the current design is optimal in performance and their corresponding benefits. Details are in the Appendix B.

$$R_u(v) = \widetilde{IG}_v(X) := (-p_v^A \log p_v^A - p_v^N \log p_v^N)/(1 + \lambda^{-1}|p_v^A - p_v^N|) \qquad (10)$$

This definition ensures that $R_u(v)$ falls within the range $[0, 1]$, providing a normalized and consistent reward to measure uncertainty reduction. The reward function reaches its maximum when the subsets $\Omega_v^A$ and $\Omega_v^N$ have equal probability, reflecting the maximum reduction in uncertainty. It reaches its minimum when one of the subsets has zero probability, indicating no reduction in uncertainty. Appendix G plots the reward function curve across values of $p_v^A$ and $p_v^N$.

## 2.5   Question Selection Via Reward Propagation

Single-step rewards often fall short in dynamic settings as they only consider immediate impact, overlooking long-term effects. To overcome this, our method uses a reward propagation scheme across simulation trees by defining 'accumulated rewards' that gather rewards over multiple simulation steps to reflect the effectiveness of past decisions. These accumulated rewards help compute 'expected rewards', indicating the likely benefits of the questions and guide the selection of candidate questions.

**Accumulated Reward**    We first define the accumulated reward at each node $v$, which accumulates the rewards at $v$ and all its ancestors on the tree, defined recursively as:

$$R_a(v) := R_u(v) + \begin{cases} 0 & v \text{ is root} \\ R_a(\mathsf{Parent}(v)) & \text{otherwise} \end{cases}$$

Here $R_u(v)$ is the uncertainty-based reward at node $v$ defined in Eq. (10), and $R_a(\mathsf{Parent}(v))$ is the accumulated reward of the parent of $v$. We compute these accumulated rewards by starting at the root and propagating down to the leaves. Intuitively, the accumulated reward at each leaf node represents the total reward we end up with at the end of the conversation at that node.

**Expected Reward**    Next, we compute the expected reward for each node $R_e(v)$, which represents the expected total value of rewards received on expectation on a node and all its descendants on tree.

$$R_e(v) := \begin{cases} R_a(v) & \text{if } v \text{ is a leaf; otherwise:} \\ p_v^A R_e(v^A) + p_v^N R_e(v^N) & \text{if } v \text{ is an Answerer Node} \\ \frac{1}{m} \sum_{w \in \mathsf{Children}(v)}^m R_e(w) & \text{if } v \text{ is a Questioner Node} \end{cases}$$

For the case where $v$ is an Answerer Node, recall that $p_v^A$ and $p_v^N$ are the conditional probabilities of affirmative and negative answers at node $v$, defined in section 2.4. $v^A$ and $v^N$ are its children, corresponding to the affirmative and negative answers. For the case where $v$ is a Questioner Node, we assign equal probability to the $m$ questions asked from this node. In this way, we propagate the expected rewards from the leaves up to the root, allowing us to compute the expected gain at the root. We compare different reward propagation schemes and find that using cumulative rewards from all paths enhances long-term decision-making benefits. See Appendix C for details.

**Determining the Optimal Question**    Finally, to decide the question to ask, we select the question with highest expected reward (and therefore, the highest expected information gain, considering both immediate and future information gains):

$$q_i = \arg\max_{n=1} R_e(q_i^n) \qquad (11)$$

## 2.6 UoT Summary

UoT first generates candidate questions $q_i^1, q_i^2, \ldots, q_i^m$ based on the history $h_i$ and current possibility set $\Omega_i$. Then, we conduct multistep simulation to generate a tree for each candidate question $q_i^n$. Next, we compute the uncertainty-based rewards $R_u(v)$, and propagate over the trees to compute accumulated reward $R_a(v)$ and expected reward $R_e(v)$. Lastly, the optimal question $q_i^n$ with highest expected reward will be selected as $q_i$ to interact with the Answerer. UoT generates candidate questions $q_i^1, q_i^2, \ldots, q_i^m$ based on history and the current possibility set $\Omega_i$. It simulates a tree for each question, calculates uncertainty-based rewards $R_u(v)$, and computes expected rewards $R_e(v)$. The question $q_i^n$ with the highest expected reward is chosen for interaction.

## 2.7 Extensions and Discussion

**Open Set UoT.** Recall that in the closed set scenario, the Questioner starts with knowledge of the possibility space $\Omega$. In practice, the possibility space is often unknown, resulting in the open set setting. To adapt UoT to this case, we prompt Questioner to initialize the possibility space $\Omega$ and then reinitialize the possibility set $\Omega_i$ according to current history $h_i$. Then, the rest of UoT is unchanged. **The generalization in open-end answers.** The UoT framework enables LLMs to update possibilities after each interaction, including affirmative/negative or open-ended responses. Thus, it can be applied to open-ended answers scenarios. **Pruned UoT.** To enhance efficiency during simulation, pruning akin to Beam Search can be employed when constructing the simulation trees, which limits the number of paths to explore over the tree to a predetermined size.

# 3 Experiments

## 3.1 Experimental Setup

**Models** We test various LLMs to evaluate the generality of our method, including **Llama-3-70B-Instruct** [1], **Mistral-Large** [21], **Gemini-1.5-Pro** [28], **Claude-3-Opus** [4] and **GPT-4** [24]. We also validate the performance of earlier released LLMs (Refer to Appendix D) including **Llama 2-70B-Chat** [32], **Cohere** [9], **PaLM 2** [2], **Claude 2** [3] and **GPT-3.5-turbo** [23].

**Baselines** **Direct Prompting (DP)** prompts an LLM directly to generate the next response. **Planning Prompting (PP)** is motivated by Wang et al.[33]. We leverage another LLM to plan the future and, consequently, determine the question to ask. **Chain-of-Thought (CoT)** [35] improves reasoning in LLMs by detailing reasoning steps. **CoT-SC (Self-Consistency)** [34] an is an ensemble method, explores multiple reasoning paths. We standardize sampling counts for fair computational cost comparison with other methods. **Reflexion** [30] lets agents propose actions and self-assess to foster new ideas. **Tree-of-Thoughts (ToT)** [38] enables LLMs to make decisions by exploring and evaluating multiple reasoning paths over a tree structure. We examine ToT under two setups: **Original-ToT**, which uses the standard approach of generating and evaluating questions, and **Adapted-ToT (Ad.-ToT)**, where we integrate heuristic experience into prompt for question generation and evaluation, focusing on questions that halve the search space. We matched the tree depth to the simulation steps in our UoT method for a fair comparison. We evaluate methods and LLMs in both open set **(OS)** and closed set **(CS)** settings. In open set, models are tested without prior knowledge of outcomes; in closed set, they are given complete information about all possible outcomes. For details, see Appendix I.1 for experimental settings and Appendix L for prompts.

**Scenarios and Datasets** **20 Questions** is a game where the *answerer* thinks of an item and the *questioner* asks up to 20 yes-or-no questions to guess it. We use two datasets, Common (collected by us, refer to Appendix I.2 for more details) and Things [14], including 111 and 1854 items separately. In this scenario, the maximal turns is set to 20. In **Medical Diagnosis**, the doctor needs to ask questions to patients about their symptoms, to determine an accurate diagnosis. We use two datasets: DX [37], with 104 doctor-patient dialogues and 5 diseases in test set, and MedDG [19] with over 17K conversations across 15 disease types. We manually selected 500 high-quality samples for evaluation (see Appendix I.3 for selection process). *Importantly, Open-ended responses from patient are allowed in MedDG to validate UoT's generalization in open-ended scenarios.* Both datasets are limited to 5 turns. **Troubleshooting** is a scenario where a customer support technician interacts with customers to identify and resolve faults or issues within computer systems, electronic devices, machinery, or

Table 1: Results from three different scenarios, assessing Success Rate (SR), Mean Conversation Length in Successful Cases (MSC), and Mean Conversation Length (MCL).

| Model | Method | 20 Questions | | | | | | Medical Diagnosis | | | | | | Troubleshooting | | |
| | | Common | | | Thing | | | DX | | | MedDG | | | FloDial | | |
| | | SR↑ | MSC↓ | MCL↓ | SR↑ | MSC↓ | MCL↓ | SR↑ | MSC↓ | MCL↓ | SR↑ | MSC↓ | MCL↓ | SR↑ | MSC↓ | MCL↓ |
|---|---|---|---|---|---|---|---|---|---|---|---|---|---|---|---|---|
| Llama3-70B | DP (OS) | 34.2 | 13.9 | 17.9 | 15.5 | 14.9 | 19.2 | 26.0 | 3.6 | 4.6 | 25.7 | 3.6 | 4.6 | 11.1 | 15.4 | 19.5 |
| | UoT(OS) | **36.9** | **12.4** | **17.3** | **21.0** | **13.6** | **18.7** | **35.6** | **2.6** | **4.1** | **50.6** | **2.3** | **3.6** | **26.1** | **9.1** | **17.2** |
| | DP (CS) | 51.4 | 14.6 | 17.2 | 15.0 | 13.8 | 19.1 | 83.7 | 3.5 | 3.7 | 60.2 | 3.5 | 4.1 | 28.8 | 15.7 | 18.8 |
| | UoT (CS) | **55.9** | **12.6** | **15.9** | **25.0** | **13.0** | **18.3** | **90.4** | **1.0** | **1.4** | **64.3** | **1.4** | **2.7** | **47.1** | **7.6** | **14.2** |
| Mistral-Large | DP(OS) | 20.7 | **13.1** | 18.6 | 12.5 | 13.6 | 19.2 | 18.3 | 3.4 | 4.7 | 28.3 | 3.2 | 4.5 | 11.1 | 15.8 | 19.5 |
| | UoT(OS) | **27.0** | 15.1 | 18.7 | **15.0** | **13.1** | **19.0** | **24.0** | **2.5** | **4.4** | **50.0** | **2.9** | **4.0** | **19.6** | **11.3** | **18.3** |
| | DP (CS) | 26.1 | 13.4 | 18.3 | 13.0 | **12.6** | 19.0 | 38.5 | 3.3 | 4.3 | 46.7 | 3.3 | 4.2 | 14.2 | 16.0 | 19.4 |
| | UoT (CS) | **31.5** | **9.8** | **16.8** | **18.5** | 13.2 | **18.7** | **48.1** | **2.2** | **3.6** | **60.0** | **1.9** | **3.2** | **30.1** | **10.9** | **17.3** |
| Gemini-1.5-Pro | DP (OS) | 36.0 | 16.8 | 18.8 | 17.5 | 14.4 | 19.0 | 26.9 | 3.5 | 4.6 | 23.7 | 4.0 | 4.8 | 9.15 | 15.6 | 19.6 |
| | UoT(OS) | **39.7** | **14.6** | **17.9** | **22.0** | **13.4** | **18.5** | **39.4** | **2.4** | **4.0** | **38.6** | **2.9** | **4.2** | **19.0** | **12.1** | **18.5** |
| | DP (CS) | 47.7 | 17.0 | 18.6 | 28.5 | 15.0 | 18.6 | 69.2 | 3.2 | 3.8 | 51.4 | 3.2 | 4.1 | 30.1 | 14.0 | 18.2 |
| | UoT (CS) | **60.4** | **13.9** | **16.3** | **32.0** | **14.0** | **18.1** | **81.7** | **2.1** | **2.6** | **81.4** | **2.1** | **2.6** | **53.6** | **11.5** | **15.4** |
| Claude-3-Opus | DP(OS) | 45.0 | **14.2** | 17.4 | 16.5 | 13.8 | 19.0 | 33.7 | 3.4 | 4.5 | 54.3 | 3.2 | 4.0 | 31.4 | 15.7 | 18.6 |
| | UoT(OS) | **63.1** | 14.4 | **16.5** | **23.5** | **13.3** | **18.4** | **45.9** | **2.6** | **3.9** | **61.5** | **2.3** | **3.3** | **35.9** | **11.0** | **16.8** |
| | DP (CS) | 52.3 | 13.8 | 16.8 | 33.5 | 14.1 | 18.0 | 75.0 | 3.3 | 3.7 | 73.3 | 3.3 | 3.8 | 48.4 | 16.0 | 18.1 |
| | UoT (CS) | **66.7** | **6.9** | **11.3** | **41.5** | **13.9** | **17.5** | **81.7** | **2.2** | **2.7** | **79.3** | **2.4** | **2.9** | **56.2** | **6.2** | **12.2** |
| GPT-4 | DP(OS) | 48.6 | **14.0** | **17.1** | 16.5 | **12.6** | 18.8 | 44.2 | 3.5 | 4.9 | 45.7 | 4.2 | 4.6 | 38.4 | 13.0 | 17.3 |
| | CoT(OS) | 13.5 | 18.6 | 19.8 | 6.00 | 16.4 | 19.0 | 18.3 | 3.8 | 4.8 | 9.71 | 4.0 | 4.9 | 30.7 | **10.3** | 17.0 |
| | Ad.-ToT(OS) | 45.0 | 17.8 | 19.0 | 21.0 | 15.2 | 19.0 | 45.2 | **2.4** | 3.8 | 51.4 | 2.7 | 3.8 | 35.3 | 13.3 | 17.7 |
| | UoT(OS) | **55.3** | 15.1 | 17.4 | **28.0** | 14.9 | **18.6** | **49.1** | **2.4** | **3.7** | **67.4** | **2.5** | **3.5** | **43.5** | 12.0 | **16.8** |
| | DP (CS) | 50.5 | 13.1 | 16.5 | 30.5 | 13.1 | 17.9 | 91.3 | 3.0 | 3.3 | 72.3 | 4.2 | 4.4 | 43.7 | 13.4 | 17.1 |
| | PP (CS) | 38.7 | 14.9 | 18.0 | 18.0 | 14.5 | 19.0 | 58.6 | 2.5 | 3.5 | 62.3 | 3.8 | 4.3 | 39.2 | 14.2 | 17.7 |
| | CoT (CS) | 20.7 | 16.0 | 19.2 | 10.0 | 16.2 | 19.6 | 33.7 | 3.7 | 4.4 | 20.0 | 3.8 | 4.3 | 32.8 | 10.1 | 16.8 |
| | CoT-SC (CS) | 55.1 | 14.0 | 16.7 | 18.5 | 14.8 | 19.0 | 48.5 | 3.6 | 4.3 | 26.7 | 4.2 | 4.8 | 42.5 | 11.0 | 16.2 |
| | Reflexion (CS) | 67.6 | 12.0 | 14.6 | 31.5 | 13.6 | 18.0 | 52.5 | 3.7 | 4.3 | 30.3 | 4.0 | 4.7 | 28.6 | 11.5 | 17.8 |
| | Original-ToT (CS) | 28.8 | 15.5 | 18.7 | 18.5 | 15.1 | 19.1 | 70.3 | 3.3 | 3.8 | 60.3 | 3.2 | 3.9 | 40.4 | 11.6 | 16.6 |
| | Ad.-ToT (CS) | 42.6 | 12.2 | 16.1 | 25.0 | **13.0** | 18.3 | 92.1 | **1.9** | 2.2 | 78.0 | 3.0 | 3.4 | 60.3 | 8.2 | 12.9 |
| | Pruned UoT (CS) | 62.2 | **10.8** | 14.3 | 34.0 | 14.9 | 18.3 | 92.1 | **1.9** | **2.1** | 83.3 | 2.7 | 3.1 | 63.2 | 8.2 | 12.5 |
| | UoT (CS) | **71.2** | **10.8** | **13.5** | **37.5** | 14.4 | **17.9** | **97.0** | 2.0 | **2.1** | **88.0** | **2.6** | **2.9** | **67.3** | **7.8** | **11.8** |

other complex systems. Raghu et al.[27] introduce FloDial with 894 dialogues, containing 153 faults and we also conduct the data preprocessing of FloDial (See Appendix I.4 for details). We evaluate using a maximum of 20 turns. The answerer, simulated by GPT-4, is prompted with the patient's actual disease and conversation details for each case. For more details, refer to Appendix I.2 and see examples of these scenarios in Appendix K.

**UoT (Open Set) Setup** We iteratively update LLMs' perceived possibilities based on conversational history, rather than defining them all upfront. In medical diagnosis and troubleshooting, initial descriptions from symptoms or issues help set up initial possibilities. In the 20-question game, we start with broad inquiries using the Direct Prompting method for the first three rounds to gather more information. The ToT tree structure method employs a similar strategy. Setup details in Appendix I.5.

**Evaluation Metrics** To measure efficacy and efficiency, we use: **Success Rate (%)**: $SR = S/T$, where $S$ is the number of successful cases, and $T$ is the total number of cases; **Mean Conversation Length in Successful Cases**: $MSC = R_s/S$, where $R_s$ is the total rounds in successful cases; **Mean Conversation Length**: $MCL = R/T$, where $R$ is the total rounds in all cases. **MCL** measures efficiency based on the resources used in both successes and failures.

### 3.2 Performance

**20 Questions** As illustrated in Table 5, for all types of LLMs, those equipped with UoT outperform the baselines in both open set and close settings. Among the methods used on GPT-4 to enhance planning and reasoning, CoT (CS) and PP (CS) show inferior performance even compared to GPT-4 alone. UoT (OS) demonstrates superior performance, with with an average 8.7% improvement than Adapted-ToT (OS) in success rate. Moreover, UoT (CS) achieves the highest success rate, surpassing the second-best Reflexion by an average of 4.3%.

**Medical Diagnosis** UoT (CS) outperforms baselines in simplified medical diagnostics, achieving a 97.0% success rate on the DX dataset with GPT-4. On the MedDG dataset, UoT (CS) on Gemini-1.5-Pro and GPT-4 achieve success rates of 81.4% and 88.0%. It also reduces conversation lengths to an average MSC of 2.0 on GPT-4 for DX, lower than 3.5 and 3.0 for DP methods. *These results demonstrate the versatility of our UoT in handling both binary and open-ended interactions effectively.*

**Troubleshooting** UoT (CS) with GPT-4 similarly achieves the highest SR of 67.3%, and the lowest MSC of 7.8. It also shows a remarkable improvement from 43.7% to 67.3% in Success Rate.

**Overall Performance** On average, UoT enhances the success rate by 38.1% compared to DP across 5 datasets and 5 different LLMs, including open source and commercial models. Notably, Success Rate increases 46.6% for Llama3-70B. Furthermore, UoT outperforms CoT-SC by 33.8% and Reflexion by 29.9%. Even compared to tree structure methods like Original-ToT and Adapted-ToT, UoT still shows superior performance with gains of 28.3% and 12.4% respectively. Additionally, Pruned UoT, our pruning method to improve efficiency, outperforms Adapted-ToT by 7.36%. Additionally, our study shows that UoT's one-step planning is effective due to effective reward design and question selection. We limit simulations to three steps for budgetary reasons, balancing efficiency and effectiveness (see Appendix E for further details on simulation depth). To determine whether the differences in success rates between the two methods were statistically significant, we performed a t-test. The results and details are in Appendix H.

**Case Studies and Reliability of GPT-4 as answerer** Figure 3 shows UoT, compared to direct prompting, more effectively reduce uncertainty and narrow down candidates, avoiding overly specific queries. After gaining initial information (e.g., stomach pain), it generates targeted questions about related issues rather than general inquiries. Additionally, GPT-4's accuracy as answerer is evaluated by analyzing 10% of interactions from each dataset, consistently showing reliable responses. For quantitative details, see Appendix F.

### 3.3 Analysis

#### 3.3.1 Comparing Model Performance at Equal Computational Efficiency

We compare the performance of approaches with similar computational costs in a closed set setting, in terms of token consumption. To do so, we first prune our UoT as described in section 2.7. Secondly, we expand exploration depth of Adapted-ToT method to bring its token cost in line with that of UoT.

As shown in the top half of Table 2, the Pruned UoT method, despite its reduced efficacy compared to UoT, still outperforms ToT and other methods. Also, the bottom part of Table 2 shows that even when increasing the depth of Adapted ToT (Adapted-ToT ($D = 4$)) to match the token cost of UoT ($D = 3$), it still underperforms compared to UoT.

#### 3.3.2 Effectiveness of Uncertainty Rewards

To further demonstrate the effectiveness of our uncertainty-based reward, we compare it with the self-evaluation reward used in the original ToT based on GPT-4 model. We implement the uncertainty-based reward in place of the self-evaluation reward in ToT, creating a variant we call ToT (+UR). The results, as shown in left side of Figure 4, indicate that our reward significantly enhances planning efficacy by an average of 5.9%. Additionally, we use the heuristic self-evaluation reward in Adapted-ToT to replace our current uncertainty-based reward in UoT, a variant we refer to as UoT (-UR). This change results in a performance decrease shown in the right part of Figure 4, further validating the effectiveness of our uncertainty-based reward. Moreover, the performance of UoT (-UR) still surpasses that of Adapted-ToT illustrated in Table 5,

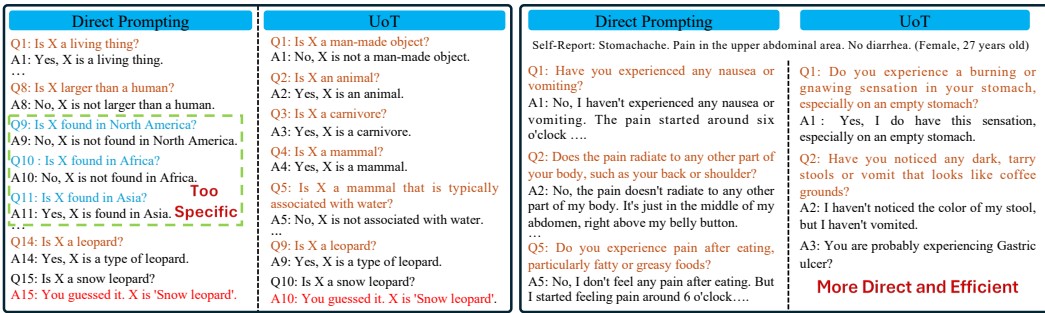

Figure 3: Case studies from the 20 Questions game (left) and simplified medical diagnosis (right).

Table 2: Average success rates for 20Q, MD, and TB at comparable efficiency, measured by GPT-4 token use. $k$ is sampling count, $D$ is tree depth.

| Method | Tokens | 20Q | MD | TB |
|---|---|---|---|---|
| CoT-SC($k = 33$) | 4.6k | 32.6 | 37.6 | 42.5 |
| Orig-ToT($D = 3$) | 4.5k | 23.7 | 65.3 | 40.4 |
| Adapt-ToT($D = 3$) | 4.5k | 33.8 | 85.1 | 60.3 |
| Pruned UoT($D = 3$) | 4.7k | **48.1** | **88.4** | **63.2** |
| Adapt-ToT($D = 4$) | 9.3k | 40.9 | 86.7 | 63.7 |
| UoT($D = 3$) | 9.2k | **54.4** | **92.5** | **66.0** |

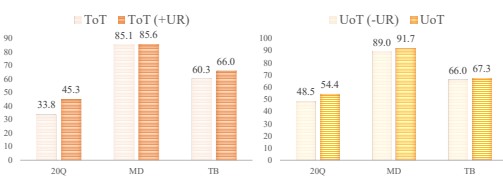

Figure 4: Success rate comparison between Adapted-ToT and Adapted-ToT using uncertainty reward, and between UoT and UoT without uncertainty reward.

## 4 Related Work

**Planning and Reasoning of LLMs** LLMs show prowess in planning and reasoning. Wei et al.[35] introduced CoT prompting for intermediate reasoning; Yao et al.[38] proposed ToT prompting using DFS/BFS. Besta et al.[6] present GoT to solve elaborate problems. Feng et al.[12] illustrated TS-LLM's tree-search guided decoding. ReAct [39] offers acting-based prompting, while Reflexion [30] enhances this with feedback reflection. Zhou et al.[41] unify reasoning and planning.

**Decision-making and Information-seeking by LLMs** LLMs have evolved as decision-making tools, with models like LLM+P [18] and LLM-DP [10] combining external planners and LLMs for natural language-based programming. RAP [13] goes beyond structured language, using LLMs with Monte Carlo Tree Search (MCTS) [7] for dynamic decision-making. This approach is also seen in the work of Zhao et al.[40], applying MCTS and LLM knowledge for complex tasks like robot control. However, MCTS struggles in uncertain scenarios due to its reliance on terminal states and specific modules for rewards and action selection. Additionally, to enhance LLMs' questioning abilities, Deng et al.[11] introduce the Rephrase and Respond method. AVIS [15] represents an autonomous visual question answering system that uses external tools. Pan et al.[25] introduce KwaiAgents for processing queries, following guidelines, and accessing external documents. Frameworks such as MEDIQ [17] and MDAgents [16] improve the reliability of LLMs in clinical settings by strengthening information-seeking capabilities and agent systems, thereby supporting more realistic diagnostic processes. [5] also explore Chatgpt's information seeking strategy in 20-questions game.

## 5 Limitation and Future Work

In practice, $\Omega_v^A$ and $\Omega_v^N$ might overlap, as different answers (such as "yes" or "no") may lead to the exclusion of different sets of possibilities. Another similar limitation is that some questions or answers may not fully eliminate certain possibilities (e.g.,"I don't have a fever" does not 100% eliminate the possibility of having COVID-19). Furthermore, compared to completely open-ended interaction in medical diagnosis or troubleshooting, our current benchmark represents a simplified scenario. In theory, such cases could be handled using the method of converting interactions into probability estimations and applying some kind of Bayesian update to the probabilities of each possibility, rather than just eliminating some subset.

## 6 Conclusion and Discussion

This paper presents the Uncertainty of Thoughts (UoT) algorithm, significantly improving LLMs in tasks requiring active information seeking through tree-based simulation, uncertainty-based rewards and a reward propagation scheme. On five datasets UoT increases success rate by 38.1% on average, establishing a new benchmark for evaluating LLMs in active information-seeking tasks. We evaluate UoT on simplified scenarios; more realistic scenarios raise challenges like allowing incomplete elimination of possibilities by answers, and others which we leave for future work.

## 7  Acknowledgment

Pang Wei Koh is supported by the Singapore National Research Foundation and the National AI Group in the Singapore Ministry of Digital Development and Innovation under the AI Visiting Professorship Programme (award number AIVP-2024-001).

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

# A Derivation of Information Gain Formula

Recall that the information gain at node $v$ is defined as the expected change in uncertainty (or entropy) when receiving an answer at this node, which we defined as:

$$IG_v(X) := H_v(X) - p_v^A \cdot H_v^Y(X) - p_v^N \cdot H_v^N(X) \tag{12}$$

We now show that:

**Proposition 1.** *The information gain at node $v$ is equal to:*

$$IG_v(X) = -p_v^A \log p_v^A - p_v^N \log p_v^N \tag{13}$$

*Proof.* Note that for any outcome $x_i$, we have by the rules of conditional probability:

$$p(x_i|\Omega_v^A) = \frac{p(x_i|\Omega_v)}{p(\Omega_v^A|\Omega_v)} = \frac{p(x_i|\Omega_v)}{p_v^A} \tag{14}$$

Now the information gain is:

$$
\begin{aligned}
IG_v&(X) \\
&= H_v(X) - p_v^A \cdot H_v^A(X) - p_v^N \cdot H_v^N(X) \\
&= - \sum_{i:\omega_i \in \Omega_v} p(x_i|\Omega_v) \log p(x_i|\Omega_v) \\
&+ p_v^A \sum_{i:\omega_i \in \Omega_v^A} p(x_i|\Omega_v^A) \log p(x_i|\Omega_v^A) \\
&+ p_v^N \sum_{i:\omega_i \in \Omega_v^N} p(x_i|\Omega_v^N) \log p(x_i|\Omega_v^N) \\
&= \sum_{i:\omega_i \in \Omega_v^A} p(x_i|\Omega_v^A)(\log p(x_i|\Omega_v^A) - \log p(x_i|\Omega_v)) \\
&+ \sum_{i:\omega_i \in \Omega_v^N} p(x_i|\Omega_v^N)(\log p(x_i|\Omega_v^N) - \log p(x_i|\Omega_v)),
\end{aligned}
$$

where the last equality holds by $p_v^A \cdot p(x_i|\Omega_v^A) = p(x_i|\Omega_v)$, and similarly for $p_v^N$. We further compute that

$$
\begin{aligned}
\sum_{i:\omega_i \in \Omega_v^A} &p(x_i|\Omega_v^A)(\log p(x_i|\Omega_v^A) - \log p(x_i|\Omega_v)) \\
&= \sum_{i:\omega_i \in \Omega_v^A} p(x_i|\Omega_v^A) \log \frac{p(x_i|\Omega_v^A)}{p(x_i|\Omega_v)} \\
&= - \sum_{i:\omega_i \in \Omega_v^A} p(x_i|\Omega_v^A) \log p_v^A \\
&= -p_v^A \log p_v^A
\end{aligned}
$$

Analogously the remaining term is $-p_v^N \log p_v^N$. Finally we conclude that

$$IG_v(X) = -p_v^A \log p_v^A - p_v^N \log p_v^N \tag{15}$$

$\square$

In fact, this proposition can also be proven using some properties of information theory, particularly the definitions of conditional entropy and mutual information. As the more computational proof shown here is still relatively short and does not require defining certain additional probability distributions, we provide the computational proof here instead.

# B   Comparison of Various Scaling Methods in Reward Function Design

We also consider multiple scaling schemes, including Logarithmic Transformation Scaling, Sigmoid Transformation Scaling and Piecewise Function Scaling. The results demonstrate that our current setting is the optimal one. Additionally, our current design, particularly setting lambda > 0, is intended as a straightforward method to incorporate our preference for a sharper reward, as it accelerates the decay of rewards as we move away from 0.5. Furthermore, it is also intended to penalize questions that are too specific when the set of possibilities remains relatively large as $|p_v^A - p_v^N|$ will be large. We elaborate all the scaling methods and their corresponding results below.

**Vanilla Expected Information Gain (IG)**

$$IG_v(X) = -p_v^A \log p_v^A - p_v^N \log p_v^N \tag{16}$$

**Logarithmic Transformation Scaling (LTS), where $k = 1$**

$$L(IG_v(X)) = \frac{\log(1 + k \cdot IG_v(X))}{\log(1 + k)} \tag{17}$$

**Sigmoid Transformation Scaling (STS), where $\tau = 10$ and $\theta = 0.5$**

$$S(IG_v(X)) = \frac{1}{1 + e^{-\tau(IG_v(X) - \theta)}} \tag{18}$$

**Piecewise Function Scaling (PFS), where $\lambda = 0.5$**

$$P(IG_v(X), p_v^A) = \begin{cases} \frac{IG_v(X)}{\lambda} \cdot p_v^A & \text{if } p_v^A \leq \lambda \\ \frac{IG_v(X)}{1-\lambda} \cdot (1 - p_v^A) & \text{if } p_v^A > \lambda \end{cases} \tag{19}$$

**Uncertainty-based Reward (UR)**

$$R_u(v) = \widetilde{IG}_v(X) := \frac{-p_v^A \log p_v^A - p_v^N \log p_v^N}{1 + \lambda^{-1}|p_v^A - p_v^N|} \tag{20}$$

In particular, in this experiment, we use 20Q-BIG-bench (introduced in §I.2) and Common dataset instead of Thing dataset in 20 Question scenario. Datasets are the same as the main chapters in other scenarios.

| Model | 20Q-BIG-bench | Common | DX | MedDG | FloDial |
|:---:|:---:|:---:|:---:|:---:|:---:|
| IG | 51.7 | 41.4 | 90.4 | 81.1 | **67.9** |
| LTS | 51.7 | 40.5 | 91.3 | 78.0 | 65.4 |
| STS | 51.3 | 35.1 | 89.4 | **82.3** | 63.4 |
| PFS | 37.9 | 36.9 | 89.4 | 81.3 | 67.1 |
| UR | **51.7** | **44.2** | **92.1** | 81.3 | 67.1 |

Table 3: Performance(Successful Rate) comparison of different reward methods based on GPT-3.5

# C   Comparison of Different Reward propagation Schemes

We also consider different reward propagation schemes and introduce their benefits as well as drawbacks.

Cumulative Reward Path Selection (CRPS): We used the strategy of calculating and comparing the cumulative reward for each path (from the root node to the leaf node), which involves multiplying the rewards of all nodes along the path and then selecting the path with the highest cumulative reward for the first question to interact with the user. This method focuses on identifying the single path that is most likely to yield a high reward. Its main limitation is that it may rely too heavily on the performance of a single path, neglecting the exploration of the overall problem space.

UoT-Max: Similar to the reward propagation scheme we are currently using, we considered adopting the approach of selecting the maximum reward among the children nodes (when the node is a questioner node) in the calculation of the expected reward. Opting for the maximum child node reward tends to pursue high rewards more aggressively, which may be more effective in some situations but could also overlook the need for exploration, potentially not always being optimal in the long run.

$$R_e(v) := \begin{cases} R_a(v) & \text{if } v \text{ is a leaf; otherwise:} \\ p_v^A R_e(v^A) + p_v^N R_e(v^N) & \text{if } v \text{ is an Answerer Node} \\ \max_{w \in \text{Children}(v)} R_e(w) & \text{if } v \text{ is a Questioner Node} \end{cases}$$

In particular, in this experiment, we use 20Q-BIG-bench (introduced in §I.2) and Common dataset instead of Thing dataset in 20 Question scenario. Datasets are the same as the main chapters in other scenarios.

| Models | Method | 20Q-BIG-bench | Common | DX | MedDG | FloDial |
|---|---|---|---|---|---|---|
| | CRPS | 62.1 | 47.7 | **92.1** | 81.3 | 56.2 |
| GPT-3.5 | UoT-Max | 48.3 | 41.4 | **92.1** | 80.3 | 60.1 |
| | UoT | **65.5** | **62.2** | **92.1** | **83.3** | **63.2** |
| | CRPS | 75.9 | 68.5 | 94.2 | 82.9 | 61.4 |
| GPT-4 | UoT-Max | 79.3 | 63.7 | 95.1 | 83.1 | 62.6 |
| | UoT | **79.3** | **71.2** | **97.0** | **88.0** | **67.3** |

Table 4: Performance (Success Rate) comparison of different reward propagation schemes. The results also demonstrate the superiority of our current reward propagation scheme.

Compared to the other two reward propagation schemes, the existing approach takes into account the cumulative rewards of all paths, providing a more holistic and balanced decision-making mechanism. Instead of merely relying on the maximum short-term rewards or the performance of a single path, it is designed to capture long-term benefits, focusing on sustainable outcomes rather than immediate short-term gains.

# D    Experimental Performance for Earlier Released LLMs

In these experiments, we use 20Q-BIG-bench (introduced in §I.2) and Common dataset instead of Thing dataset in 20 Question scenario. Datasets are the same as the main chapters in other scenarios.

Table 5: Results from three different scenarios, assessing Success Rate (SR), Mean Conversation Length in Successful Cases (MSC), and Mean Conversation Length (MCL).

| Model | Method | 20Q in BIG-bench | | | Common | | | DX | | | MedDG | | | FloDial | | |
|---|---|---|---|---|---|---|---|---|---|---|---|---|---|---|---|---|
| | | SR↑ | MSC↓ | MCL↓ | SR↑ | MSC↓ | MCL↓ | SR↑ | MSC↓ | MCL↓ | SR↑ | MSC↓ | MCL↓ | SR↑ | MSC↓ | MCL↓ |
| Llama2-70B | DP(OS) | 6.90 | **12.0** | 19.5 | 1.80 | **11.0** | 19.8 | 13.4 | 3.1 | 4.8 | 23.7 | 3.4 | 4.6 | 11.1 | 15.1 | 19.5 |
| | DP(CS) | 17.2 | 13.5 | 18.9 | 6.31 | 12.0 | 19.7 | 29.8 | 3.0 | 4.4 | 28.0 | 3.5 | 4.6 | 24.2 | **14.5** | 18.7 |
| | UoT(CS) | **20.7** | 13.2 | **18.6** | **10.8** | 15.6 | **19.5** | **51.9** | **1.8** | **3.4** | **33.9** | **1.4** | **3.8** | **31.4** | 15.8 | 18.7 |
| Cohere | DP(OS) | 3.45 | 15.0 | 19.8 | 1.80 | 14.0 | 19.9 | 19.8 | 3.7 | 4.7 | 25.0 | 3.6 | 4.7 | 16.3 | 16.7 | 19.5 |
| | DP(CS) | 6.90 | 12.0 | 19.4 | 1.80 | 12.5 | 19.8 | 35.6 | 3.3 | 4.4 | 33.3 | 4.0 | 4.7 | 27.5 | 16.3 | 19.0 |
| | UoT(CS) | **34.3** | **8.50** | **16.0** | **16.2** | **11.7** | **18.6** | **45.5** | **2.6** | **3.9** | **75.7** | **2.7** | **3.3** | **41.4** | **8.7** | **15.3** |
| PaLM 2 | DP(OS) | 37.9 | 13.5 | 17.5 | 35.1 | 14.4 | 18.0 | 7.69 | 3.9 | 4.9 | 11.3 | 4.0 | 4.9 | 22.6 | 15.2 | 19.0 |
| | DP(CS) | 51.7 | 13.2 | 16.5 | 53.1 | 13.9 | 16.8 | 7.92 | 3.4 | 4.9 | 34.0 | 4.4 | 4.8 | 30.1 | 15.0 | 18.5 |
| | UoT(CS) | **72.4** | **7.0** | **10.6** | **62.1** | **12.5** | **15.3** | **75.0** | **2.1** | **2.8** | **80.7** | **2.2** | **2.7** | **48.4** | **7.6** | **14.0** |
| Gemini-1.0-Pro | DP(OS) | 10.3 | 8.3 | 18.8 | 11.7 | 10.0 | 18.8 | 12.5 | 3.2 | 4.8 | 30.7 | 3.7 | 4.6 | 2.61 | 13.0 | 19.8 |
| | DP(CS) | 20.7 | 14.8 | 18.9 | 12.6 | 12.0 | 19.0 | 64.4 | 3.3 | 3.9 | 40.7 | 3.5 | 4.4 | 5.23 | 16.1 | 19.8 |
| | UoT(CS) | **31.0** | **7.8** | **16.2** | **18.9** | **4.0** | **17.0** | **67.3** | **2.1** | **3.7** | **75.0** | **1.4** | **2.7** | **14.2** | **10.6** | **18.6** |
| Claude2 | DP(OS) | 48.3 | 9.8 | 15.1 | 29.7 | 13.8 | 18.2 | 45.2 | 3.0 | 4.1 | 60.7 | 4.1 | 4.5 | 39.7 | 14.3 | 17.7 |
| | DP(CS) | 72.4 | 11.6 | 13.9 | 43.2 | 13.8 | 17.3 | 97.1 | 2.4 | 2.5 | 83.0 | 4.3 | 4.4 | 42.9 | 15.7 | 18.2 |
| | UoT | **75.9** | **5.1** | **8.69** | **61.3** | **9.8** | **13.7** | **98.0** | 2.3 | **2.4** | **88.3** | 2.7 | **2.9** | **52.6** | **6.3** | **12.8** |
| GPT-3.5 | DP(OS) | 36.0 | 12.6 | 17.3 | 32.6 | 14.6 | 18.2 | 18.8 | 3.5 | 4.7 | 25.0 | 3.5 | 4.6 | 19.4 | 12.3 | 18.5 |
| | UoT(OS) | 41.4 | 13.8 | 17.4 | 34.2 | 14.7 | 18.2 | 37.5 | 2.4 | 4.0 | 61.0 | 2.3 | 3.3 | 26.1 | 11.3 | 17.7 |
| | DP(CS) | 44.8 | 13.2 | 17.0 | 40.0 | 14.8 | 17.8 | 49.5 | 2.7 | 3.3 | 42.3 | 3.8 | 4.5 | 22.6 | 13.3 | 18.5 |
| | UoT(CS) | **51.7** | **5.3** | **12.4** | **44.2** | **10.9** | **16.0** | **92.1** | **2.1** | **2.4** | **81.3** | **2.4** | **2.9** | **67.1** | **6.9** | **11.2** |

# E  Effect of Simulation Depth

In this experiment, we use 20Q-BIG-bench (introduced in §I.2) and Common dataset instead of Thing dataset in 20 Question scenario. Datasets are the same as the main chapters in other scenarios.

As the below figure illustrates, we analyze the impact of simulation steps. Even with one-step reasoning and planning, our method can still have a strong performance, further indicating the effectiveness of our reward design and question selection mechanism. With the increase of the step, the performance can gradually rise. However, due to the constraints of computation resources and OpenAI API budgets, we only explore the simulation to the third step and argue that it can be the practical tradeoff between performance and efficiency.

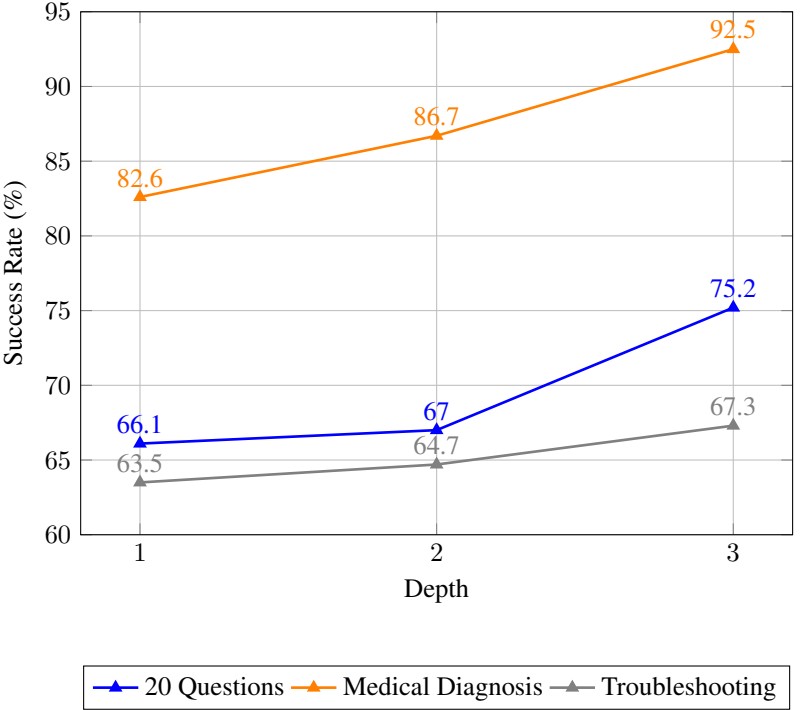

# F  Reliability of GPT-4 as the Environment

As the impressive understanding ability of LLMs, previous research has validated the effectiveness of evaluators served by ChatGPT or GPT-4 [8, 20]. Consequently, we also adopt GPT-4 as the environment to provide feedback on our work. Prompts can be found in Appendix L.4. To assess the accuracy and reliability of employing GPT-4 as the environment simulator, we randomly sample 10% interaction records (including the final judgment and intermediate feedback from the environment) from each dataset. As Figure 6 shows, GPT-4 can provide completely accurate judgment and also keep a high level of accurate feedback during the interaction. These experimental results can further support the effectiveness of our method.

Table 6: Human evaluation results for the accuracy of environment feedback served by GPT-4. IF represent the Accuracy of **I**ntermediate **F**eedback.

| Scenario | Judgement | IF |
|---|---|---|
| 20 Questions | 100 | 93.7 |
| Medical Diagnosis | 100 | 94.4 |
| Troubleshooting | 100 | 92.9 |

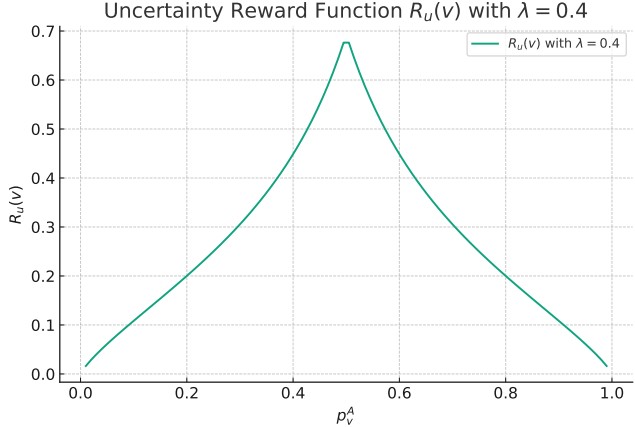

Figure 5: Curve of uncertainty based reward on Eq 10, where $p_v^N$ can be replaced by $(1 - p_v^A)$. The horizontal axis $p_v^A$ is conditional probabilities of affirmative at node v, which are introduced in Section §2.4.

## G   Reward Function Details and Its Curve

Refer to Figure 5 for the curve of uncertainty-based reward function.

## H   Experimental Statistical Significance

We conduct three experiments on five datasets using Llama 3 and GPT-4 to compare the performance of Direct Prompting (DP) and UoT methods in a closed-set setting for significance testing. Due to LLM API quota limitations, the number of experiments are restricted. To determine whether the differences in success rates (SR) between the two methods were statistically significant, we performed a t-test. The results are presented below.

**GPT-4 Results**

Table 7: GPT-4 Comparison of DP and UoT on Success Rates (SR)

| Dataset | DP (%) | UoT (%) | t-Statistic | p-Value | Significance Conclusion |
|---------|--------|---------|-------------|---------|-------------------------|
| Common  | 49.0   | 70.9    | -10.8       | 0.00041 | Significant ($p < 0.05$) |
| Thing   | 30.8   | 36.8    | -8.04       | 0.00129 | Significant ($p < 0.05$) |
| DX      | 89.4   | 97.0    | -3.11       | 0.03581 | Significant ($p < 0.05$) |
| MedDG   | 74.9   | 87.9    | -7.33       | 0.00185 | Significant ($p < 0.05$) |
| FloDial | 42.5   | 67.8    | -19.8       | 0.00004 | Significant ($p < 0.05$) |

**Llama 3 Results**

Table 8: Llama 3 Comparison of DP and UoT on Success Rates (SR)

| Dataset | DP (%) | UoT (%) | t-Statistic | p-Value | Significance Conclusion |
|---------|--------|---------|-------------|---------|-------------------------|
| Common  | 47.7   | 56.5    | -4.39       | 0.01180 | Significant ($p < 0.05$) |
| Thing   | 14.8   | 24.8    | -16.0       | 0.00009 | Significant ($p < 0.05$) |
| DX      | 80.1   | 90.1    | -4.65       | 0.00966 | Significant ($p < 0.05$) |
| MedDG   | 61.3   | 64.6    | -4.15       | 0.01426 | Significant ($p < 0.05$) |
| FloDial | 29.9   | 46.4    | -10.5       | 0.00047 | Significant ($p < 0.05$) |

The t-test results indicate that UoT significantly outperform DP five datasets (p < 0.05), as evidenced by their higher mean scores.

# I Experimental Setups

## I.1 Baselines Setup

**Chain-of-Thought (CoT)**     We adapt the typical CoT prompt which instruct LLM to generate the explanation or motivation for the proposed question first, then give the question to ask.

**Chain of Thought with Self-Consistency (CoT-SC)**     To make the method spend comparable compute to our approach for a fair comparison, we sampled 33 times before deciding on each action with the LLM's temperature of 0.7. The final selected question is the one repeated most times among 33 samples.

**Planning Prompting**     To measure whether LLMs' planning ability can be enhanced through some crafted prompts like CoT, ToT or Reflexion. We design the prompt to enable LLM to simulate multiple different sets of future interactions between questioner and answerer, then let LLM choose one most promising interaction (question) to ask.

**Tree of Thoughts**     In the case of **Original-ToT**, a sampling method is employed to generate 3 questions from each answer node, and the self-evaluation method is utilized for reward calculation. Subsequently, breadth-first search will be used and 10 nodes from each step will be selected for later simulation. Additionally, the temperature of the LLM is configured to 0.7, consistent with the settings in original ToT paper. In the case of **Adapted-ToT**, we provide more heuristical hints in prompt to generate the questions, e.g. 'you should try to propose the question to halve the probability set'. Likewise, each answer node generates 3 questions, and the LLM selects 10 nodes with higher self-evaluation rewards to further simulation. The simulation steps are also 3.

**Reflextion**     This approach involves the LLM agent suggesting questions iteratively until the question reward exceeds the threshold of 0.7 or reaches the maximum limit of 3 questions. The reward score $s$ is calculated using the formula $s = \min(p^A, p^N)/\max(p^A, p^N)$. This heuristic is based on the principle of whether the question can effectively halve the probability set. If a candidate question achieves a score above the threshold, the process of proposing questions is concluded, and that question is selected. In cases where no question meets the threshold, the one with the highest score is chosen.

**Uncertainty of Thoughts Pruned**     After generating the candidate question based on the possibility set $\Omega_i$, we sorted these question nodes by uncertainty based reward and reserved half of them, serving the purpose of pruning. In subsequent steps of the simulation, this pruning operation will be continued. Other settings were the same as UoT, described in Section §I.6.

## I.2 Scenarios Settings and Datasets

**20 Questions game** is a classic guessing game where the *answerer* thinks of an object, person, place, or other, and the *questioner*, possessing no prior knowledge about the chosen entity, proceeds to pose a series of up to 20 yes-or-no questions to determine what the secret item is. The questions are designed to narrow the possibilities and ultimately guess the secret item within the 20 questions. **20 Questions in BIG-bench**: It is the sub-task of BIG-bench and can be found on the GitHub website[7], consist of 29 items. **Common Dataset Construction**: We came across an official website[8] that introduces a 20 Questions game, which mentions that common target categories in this game include animals, places, food, and objects. Therefore, we extracted and manually screened the targets mentioned on this website, resulting in a dataset named "Common" comprising 111 targets, each belonging to one of the four aforementioned categories. **Thing Dataset**: It is a collection of 1,854 varied object concepts, carefully selected from tangible and easily identifiable nouns in American English by Martin at al.[14], which is publicly available on their official website[9].

---

[7]https://github.com/google/BIG-bench/tree/main/bigbench/benchmark_tasks/twenty_questions
[8]https://blog.prepscholar.com/20-questions-game
[9]https://osf.io/jum2f

**Medical Diagnosis**     In this scenario, the patient will simply describe their symptom first which we call a 'Self-report', then doctor acted by LLM will start to ask questions to interact with patient to determine the disease.

**Troubleshooting**     In FloDial dataset, trouble includes faults of car and laptop. Similar to Medical Diagnosis, the customer first describes some simple fault symptoms, then the customer support technician will chat with customer to further check the specific issues of device.

**LLMs Serve as Questioner (Patient or Customer)**     In simulated interactions involving questioner and answerer scenarios, particularly for medical diagnosis and troubleshooting, the response given by an LLM acting answerer is guided by scenario instructions and real-world dialogue examples. This approach makes the responses of answerer more human-like and enhances its accuracy in diagnosing diseases or identifying faults. While, in the game of 20-question, where the objective is to guess common items, the LLM acting as the answerer only needs to provide simple 'yes' or 'no' answers. Therefore, incorporating real-world dialogue into the LLM's prompts for this game is not necessary.

### I.3   Dataset selection criteria and process for MedDG

In the original MedDG dataset, numerous conversations lacked a clear diagnosis, often concluding with advice for the patient to rest or seek further tests. This ambiguity arose from patients not detailing their symptoms sufficiently or doctors lacking the information or confidence to diagnose. Consequently, these conversations hinder LLMs from accurately understanding disease and symptom information for effective patient role simulation. To address this, we curated our final evaluation set to include only conversations with explicit disease diagnoses.

Furthermore, to ensure a balanced representation across the 8 disease categories, we selected roughly 40 dialogues for each disease. We also excluded conversations that were too brief (1-2 turns) or excessively lengthy (over 10 turns). The curation process involved two annotators: one for initial selection and another for verification.

Given these criteria, we finally pick 500 conversations for our evaluation set, aiming to maintain the evaluation's reliability and quality. We will also clarify this and add the details into the following version.

### I.4   Data Preprocessing of FloDial

We process the dataset, FloDial, to convert troubleshooting flowcharts into a set of troubleshooting faults. The dialogue is grounded in specific faults, which correspond to the leaf nodes (descriptions and solutions of faults) in the flowcharts. After reviewing all the leaf nodes, we identify 153 faults that had corresponding dialogues. We then use GPT-4 to generate a clear name for each fault based on the descriptions and solutions of faults, and randomly selected one corresponding dialogue history to construct the current dataset.

### I.5   UoT (Open Set) Setup

To initialize the possibility set as the start of the algorithm, in medical diagnosis and troubleshooting, initial descriptions from patients or customers about symptoms or issues enable UoT to establish a possibility set right from the start. For the game of 20 Questions, where initial information is scant, prematurely establishing this set could misdirect the inquiry. Therefore, for the first three rounds, we employ Direct Prompting in Open-Set (DPOS) approach to gather information and feedback. After these initial rounds, UoT takes over, refreshing the possibility set each round to refine the questioning strategy.

For datasets Common, Things, DX, MedDG and FloDial, we configure the size of the possibility set for each update round, setting them at 10, 10, 5, 5, and 5, respectively. This parameter should prevent the increase in cognitive load and decrease in efficiency that comes with larger sizes, while also avoiding the limitations of focusing on a few specific items that come with smaller sizes. We experiments with values(size) between 5 and 50 based on this rationale, and the final selection of these hyperparameters is guided by empirical performance evaluations.

## I.6 Implementation

Empirically, we set the plan (simulation) steps as 3 and the number of questions during the simulation is 3. The hyperparameter $\lambda$ in uncertainty-based reward is 0.4.

## J Impact Statement

This paper aims to enhance LLMs' information seeking abilities, allowing them to make better decisions in challenging real-world settings involving uncertainty and ambiguity, and to interact more effectively with humans in human-AI settings. On the whole, we expect that information seeking capabilities should allow models to behave in a manner that is more reliable and well-aligned with human expectations, as it allows them to better resolve their uncertainty. There may be some potential societal consequences related to LLMs in general, such as privacy and trustworthiness issues. However, we do not believe that any of these must be specifically highlighted here.

## K Examples in Scenarios

Some examples of conversation in different scenarios are provided in Table 9-12.

Table 9: Example for 20 Question. **Q** stands for Questioner and **A** stands for Answerer.

| **Target:** Glass micropipette | |
|---|---|
| **Conversation** | |
| 1: **Q:** Is X a tangible object? | **A:** Yes. |
| 2: **Q:** Is X a living organism? | **A:** No. |
| 3: **Q:** Is X a household item? | **A:** No. |
| 4: **Q:** Is X a type of transportation? | **A:** No. |
| 5: **Q:** Is X used in construction? | **A:** No. |
| 6: **Q:** Is X a wearable item? | **A:** No. |
| 7: **Q:** Is X a type of toy? | **A:** No. |
| 8: **Q:** Is X a 'Glass micropipette'? | |
|    **A:** You guessed it. X is a 'Glass micropipette'. | |

Table 10: Example for Medical Diagnosis. **D** stands for Doctor (Questioner) and **P** stands for Patient (P).

| **Disease:** Allergic rhinitis |
|---|
| **Self-report** |
| Hello doctor, my son's throat easily becomes red and swollen as soon as winter comes. He has been coughing and his throat has been red and swollen for the past half month. |
| **Conversation** |
| 1: **D:** Do you experience a runny or stuffy nose? **P:** Yes. |
| 2: **D:** You may have 'Allergic rhinitis'. |

Table 11: Example for MedDG. **D** stands for Doctor (Questioner) and **P** stands for Patient (Answerer).

| **Disease:** Gastric ulcer |
|---|
| **Self-report** |
| Stomachache. Pain in the upper abdominal area. No diarrhea. (Female, 27 years old) |
| **Conversation** |
| 1: **D:** Do you experience a burning or gnawing sensation in your stomach, especially on an empty stomach? |
|    **P:** Yes, I do have this sensation, especially on an empty stomach. |
| 2: **D:** Have you noticed any dark, tarry stools or vomit that looks like coffee grounds? |
|    **P:** I haven't noticed the color of my stool, but I haven't vomited. |
| 3: **D:** You may have 'Gastric ulcer'. |

Table 12: Example for Troubleshooting. **T** stands for customer support technician and **C** stands for customer.

| Fault: Brake fluid overflow |
| --- |
| **Self-report** |
| Lately, I have been having trouble with the brakes of my car. I have checked it again and again but can't figure out what's causing this. Please help me. |
| **Conversation** |

| | |
| --- | --- |
| 2: **T:** Does the issue affect the rotation or movement of the wheels? | **C:** No. |
| 3: **T:** Does the issue seem to be related to the overall braking system? | **C:** Yes. |
| 4: **T:** Are you experiencing a brake fluid leak? | **C:** No. |
| 5: **T:** Are your brake pads worn out? | **C:** No. |
| 6: **T:** Is the brake pedal feeling spongy or soft when you press it? | **C:** Yes. |
| 7: **T:** Is there an issue with the brake lines being air-filled or damaged? | **C:** No. |
| 8: **T:** Your car may have a brake fluid overflow issue. | |

# L  Prompts

## L.1  DP(OS) and DP(CS) Prompt

The specific prompts for the DP(OS) and DP(CS) baselines in each scenario are provided in Table 13-15.

Table 13: DPOS and DPCS Prompt for 20 Question game. The two settings are the same except for the content noted in red, which is only for DPCS Prompt. The first time the content marked in red is mentioned at the beginning of interaction is to inform the questioner of the entire probability set. The second mention serves as a reminder for the questioner to determine the target as soon as possible when the conversation is nearing its end. We establish two phases with the aim of encouraging the questioner to guess the target as soon as possible when the second phase of interaction is nearing its conclusion, in order to avoid failure at the end of interaction.

| **Prologue** |
| --- |
| Let us play the game of 20 questions. I am impersonating the thing, X. You will ask me up to 20 questions that start with 'Is X' and can only be answered by yes or no, and I will answer each one truthfully based on being X. |
| Let us begin. Ask me the first question. Reply to me with your first question. |
| The X may be one of the items below: [list of items] |
| **Phase 1** (Q1 - Q13) |
| *None* |
| **Phase 2** (Q14 - Q20) |
| The X may be one of the items below: [list of items] |

## L.2  Planning Prompt

The specific prompts for Planning Prompt baselines in each scenario are provided in Table 28-27. As planning prompt method is close set setting, hence the probability set will also be informed in the prompt as DPCS prompt. We do not repeat it in the tables.

## L.3  UoT Prompt

The detailed prompts for our UoT method in each scenario are attached in Table 19-21.

## L.4  Questioner Prompt

The specific prompt to guide the examiner in all scenarios is provided in Table 22-24.

Table 14: DPOS and DPCS are medical diagnostic prompts with similar structures. The key difference is the inclusion of probability set information in red text for DPCS. This red text appears twice: initially to inform the questioner about all potential diagnoses and again towards the end of the interaction as a reminder to quickly confirm the disease. We establish two phases with the aim of encouraging the questioner to confirm the disease as soon as possible when the second phase of interaction is nearing its conclusion, avoiding failure at the end of interaction.

| **Prologue** |
| --- |
| You are a doctor and your patient reports that: [self-report] |
| You should ask your patient questions in English with symptoms which can only be answered by yes or no, to find what disease this patient suffers. Let us begin. Ask me the first question. |
| Based on the symptoms above, if you find out the disease, please ask 'Are you a [disease name]?' The patient may suffer from one of the diseases below: [list of diseases] |
| **Phase 1** (Q1) |
| *None* |
| **Phase 2** (Q3 - Q5) |
| Based on the symptoms above, if you find out the disease, please ask 'Are you a [disease name]?' The patient may suffer from one of the diseases below: [list of diseases] |

Table 15: DPOS and DPCS are troubleshooting prompts with similar structures, but DPCS includes unique content highlighted in red. This red content appears first at the beginning, outlining all potential faults, and again towards the end as a reminder to swiftly identify the fault. The two-phase structure of these prompts aims to ensure quick fault confirmation, especially in the final stages of the interaction, to prevent failure.

| **Prologue** |
| --- |
| You are a technician and your client reports that: [self-report] |
| You should ask your client questions about a specific situation which can only be answered by yes or no, in order to find where the issue this client faces with located. Let us begin. Ask me the first question. |
| The client may face one of the issues below: [list of issues] |
| **Phase 1** (Q1 - Q13) |
| *None* |
| **Phase 2** (Q14 - Q20) |
| Based on the situations above, if you find out the issue, please ask 'Are you a [issue name]?' The client may face one of the issues below: [list of issues] |

Table 16: Planning Prompt for 20 Question game. [C1] is the count of questions asked and [C2] is the count of questions remaining. The 'information gained' marked blue represents the previous interaction history. We divide it into three phases to discuss the probability set as quickly as possible, conduct simulation for planning, and remind the questioner to guess the answer.

| **Prologue** |
| --- |
| *Same as prompts in Appendix L.1* |
| **Phase 1** (Q1 - Q4) |
| The next question should narrow down the possible range of X, preferably in half. |
| **Phase 2** (Q5 - Q15) |
| We are playing the 20 Question game, [C1] questions have been asked. And now we know: [information gained] |
| Based on the features of X above, please guess what X exactly is and tell me your top 3 most likely answers. |
| For these three candidate X, please separately complete the remaining [C2] questions and answer yes/no by yourself. Notably, you must guess the corresponding X before the last question. |
| **Phase 3** (Q16 - Q20) |
| Note that you should guess what X exactly is from now on. The question must start with 'Is X ...' |

Table 17: Planning Prompt for Medical Diagnosis. [C1] is the count of questions asked and [C2] is the count of questions remaining. The 'information gained' marked blue represents the previous interaction history. We divide it into three phases to discuss the probability set as quickly as possible, conduct simulation for planning, and remind the questioner to confirm the disease.

| **Prologue** |
| --- |
| *Same as prompts in Appendix L.1* |
| **Phase 1** |
| *Skip because of the limited QA rounds in this scenario* |
| **Phase 2** (Q1 - Q3) |
| You are the doctor asking questions to diagnose, [C1] questions have been asked. And now we know about the patient: [information gained] |
| Based on the symptoms of the patient above, please think about what disease the patient suffers from and tell me your top three most likely answers. |
| For these three candidate diseases, please separately complete the remaining [C2] questions and answer yes/no by yourself. Notably, you must determine the corresponding disease before the last question. |
| **Phase 3** (Q4 - Q5) |
| Note that you should determine what disease the patient suffers from now. The question must start with 'Are you a [disease name]?' |

Table 18: Planning Prompt for Troubleshooting. [C1] is the count of questions asked and [C2] is the count of questions remaining. The 'information gained' marked blue represents the previous interaction history. We divide it into three phases to discuss the probability set as quickly as possible, conduct simulation for planning, and remind the questioner to confirm the fault.

| **Prologue** |
| --- |
| *Same as prompts in Appendix L.1* |
| **Phase 1** (Q1 - Q4) |
| The next question should narrow down the possible range of trouble issues, preferably in half |
| **Phase 2** (Q5 - Q15) |
| You are a technician to troubleshoot, [C1] questions have been asked. And now we know: [information gained] |
| Based on the situation your client faces, please think about what the issue exactly is and tell me your top 3 most likely answers. |
| For these three candidate issues, please separately complete the remaining [C2] questions and answer yes/no by yourself. Notably, you must determine the corresponding issue before the last question. |
| **Phase 3** (Q16 - Q20) |
| Note that you should determine what issue your client faces from now on. The question must start with 'Are you a [issue name]?' |

Table 19: UoT Prompt for the 20 Questions Game: As it is based on a closed-set setting, information about probabilities will be given at the beginning of the interaction and will be reminded after Q14. Since it is similar to previous prompts, we will not repeat it here. In the 'Prompt for Question Generation and Simulation', the count of YES/NO indicates the number of items that are consistent with the affirmative/negative response.

| **Prologue** |
| --- |
| Let us play the game of 20 questions. I am impersonating the thing, X. You will ask me up to 20 questions that start with 'Is X' and can only be answered by yes or no, and I will answer each one truthfully based on being X. |
| Let us begin. Ask me the first question. Reply to me with your first question. |
| **Prompt for Question Generation and Simulation** |
| Please design a question about X and can only be answered by YES or NO. asked Then classify the possible X above based on this question. If the answer is 'YES', put this X into 'YES: ...', otherwise to 'NO: ...'. Finally calculate how many X in YES and NO. |
| Notably, this question should fulfill that the count of YES and NO are almost the same with a permissible discrepancy of no more than one! |
| You should think about best n questions to respond to. And your answer should be: |
| Question 1: Is X ...? |
| YES: item1, item2, ... |
| Count of YES: ... |
| NO: item1, item2, ... |
| Count of NO: ... |
| **Additional Reminder in Q14 - Q20** |
| Note that you should guess and ask what X exactly is from now on. X is possible a: [item_list_str], or other. The question must start with 'Is X ... |

Table 20: UoT Prompt for medical diagnosis: As it is based on a closed-set setting, information about probabilities will be given at the beginning of the interaction and will be reminded after Q3. Since it is similar to previous prompts, we will not repeat it here. In the 'Prompt for Question Generation and Simulation', the count of YES/NO indicates the number of diseases that are consistent with the affirmative/negative response.

| |
|---|
| **Prologue** |
| You are a doctor and your patient reports that: [self-report] |
| You should ask your patient questions in English with symptoms which can only be answered by yes or no, to find what disease this patient suffers. |
| Let us begin. Ask me the first question. |
| **Prompt for Question Generation and Simulation** |
| Please design a question to ask your patient with symptoms about disease and can only be answered by YES or NO. Then classify the possible disease above based on each question. If the answer is 'YES', put this disease into 'YES: ...', otherwise to 'NO: ...'. Finally calculate how many X in YES and NO. |
| Notably, this question should fulfill that the count of YES and NO are almost the same with a permissible discrepancy of no more than one! |
| You should think about best n questions to respond to. And your answer should be: Question 1: ...? YES: disease1, disease2, ... (disease names only) Count of YES: ... NO: disease1, disease2, ... (disease names only) Count of NO: ... |
| **Additional Reminder in Q3 - Q5** |
| Note that you should point out and ask what disease the patient suffers from now. The patient may suffer from one of diseases below: [list of disease], or other. The question must be 'You may have a [disease name]?' |

Table 21: UoT Prompt for troubleshooting: As it is based on a closed-set setting, information about probabilities will be given at the beginning of the interaction and will be reminded after Q14. Since it is similar to previous prompts, we will not repeat it here. In the 'Prompt for Question Generation and Simulation', the count of YES/NO indicates the number of faults which are consistent with the affirmative/negative response.

| |
|---|
| **Prologue** |
| You are a technician and your client reports that: [self-report] |
| You should ask your client questions about a specific situation which can only be answered by yes or no, in order to find where the issue this client faces with located. |
| Let us begin. Ask me the first question. |
| **Prompt for Question Generation and Simulation** |
| Please design a question to ask your client with specific situation and can only be answered by YES or NO. asked Then classify the possible issue above based on this question. If the answer is 'YES', put this issue into 'YES: ...', otherwise to 'NO: ...'. Finally calculate how many issues in YES and NO.

Notably, this question should fulfill that the count of YES and NO are almost the same with a permissible discrepancy of no more than one!

You should think about best n questions to respond to. And your answer should be:
Question 1: ...?
YES: issue1, issue2, ... (issue names only)
Count of YES: ...
NO: issue1, issue2, ... (issue names only)
Count of NO: ... |
| **Additional Reminder in Q14 - Q20** |
| Note that you should point out and ask what disease the patient suffers from now. The patient may suffer from one of diseases below: [list of disease], or other. The question must be 'You may have a [disease name]?' |

Table 22: Prompt for Answerer in 20 questions game. [target] is the name of the final target for each dialogue.

| |
|---|
| **Prologue** |
| Let us play the game of 20 questions. You are the answerer and I am the guesser. X is '[target]'. I will ask you up to 20 questions and you should answer each one truthfully based on being X. If I guess correctly what is X, answer me "You guessed it. X is '[target]'." Note that never directly tell me what X is all the time. |
| Let us begin. Here is my first question. |

Table 23: Prompt for Answerer in medical diagnosis. [disease] is the name of the final disease for each dialogue.

| |
|---|
| **Prologue** |
| You are the patient suffering '[target]' and I am the doctor. I will ask you up to 5 questions and you should answer each one truthfully based on your disease. If I point out correctly what disease you experience, answer me "You are right. I am experiencing '[target]'." Note that never directly tell me what disease is all the time. |
| Let us begin. Here is my first question. |

Table 24: Answerer Prompt in troubleshooting. [fault] is the name of the final fault for each dialogue.

| |
|---|
| **Prologue** |
| You are the client with a device that has '[target]' and I am the technician. I will ask you up to 20 questions and you should answer each one truthfully based on the issue of your device. If I point out correctly what your issue is, answer me "You are right. My device has '[target]'." Note that never directly tell me what the issue is all the time. |
| Let us begin. Here is my first question. |

Table 25: ToT Prompt for 20 Question game. [C1] is the count of questions asked. The 'information gained' marked blue represents the previous interaction history.

| Standard Prompt |
|---|
| You are playing the game of 20 questions. I am impersonating the thing, X. You will ask me up to 20 questions that start with 'Is X' and can only be answered by yes or no, and I will answer each one truthfully based on being X. [C1] questions have been asked. And now we know: [information gained] Design a question about X and can only be answer by YES or NO. |
| **Additional Reminder in Q14 - Q20** |
| *Same as prompts in Appendix L.3* |

Table 26: ToT Prompt for Medical Diagnosis. [C1] is the count of questions asked. The 'information gained' marked blue represents the previous interaction history.

| Standard Prompt |
|---|
| You are a doctor and your patient reports that: [self-report] You should ask your patient questions in English with symptoms which can only be answered by yes or no, to find what disease this patient suffers. [C1] questions have been asked. And now we know: [information gained] Design a question to ask your patient with symptoms about disease and can only be answered by YES or NO. |
| **Additional Reminder in Q14 - Q20** |
| *Same as prompts in Appendix L.3* |

Table 27: ToT prompt for Troubleshooting. [C1] is the count of questions asked. The 'information gained' marked blue represents the previous interaction history.

| Standard Prompt |
|---|
| You are a technician and your client reports that: [self-report] You should ask your client questions about a specific situation which can only be answered by yes or no, in order to find where the issue this client faces with located. [C1] questions have been asked. And now we know: [information gained] Design a question to ask your client with specific situation and can only be answered by YES or NO. |
| **Additional Reminder in Q14 - Q20** |
| *Same as prompts in Appendix L.3* |


Table 28: CoT Prompts.

| |
|---|
| **Prologue** |
| *Same as prompts in Appendix L.1* |
| **Prompt for Generating Question and Explanation** |
| What's your next question? Let's think step-by-step and reply me with your explanation. |
| Your answer should be: |
| Explanation: [insert step-by-step analysis here] |
| Question: [next question] |
| **Additional Reminder in Q14 - Q20** |
| *Same as prompts in Appendix L.3* |

**The checklist answers are an integral part of your paper submission.** They are visible to the reviewers, area chairs, senior area chairs, and ethics reviewers. You will be asked to also include it (after eventual revisions) with the final version of your paper, and its final version will be published with the paper.

The reviewers of your paper will be asked to use the checklist as one of the factors in their evaluation. While "[Yes] " is generally preferable to "[No] ", it is perfectly acceptable to answer "[No] " provided a proper justification is given (e.g., "error bars are not reported because it would be too computationally expensive" or "we were unable to find the license for the dataset we used"). In general, answering "[No] " or "[NA] " is not grounds for rejection. While the questions are phrased in a binary way, we acknowledge that the true answer is often more nuanced, so please just use your best judgment and write a justification to elaborate. All supporting evidence can appear either in the main paper or the supplemental material, provided in appendix. If you answer [Yes]  to a question, in the justification please point to the section(s) where related material for the question can be found.

IMPORTANT, please:

- **Delete this instruction block, but keep the section heading "NeurIPS paper checklist",**
- **Keep the checklist subsection headings, questions/answers and guidelines below.**
- **Do not modify the questions and only use the provided macros for your answers.**

