# OpenReview forum: "Uncertainty of Thoughts: Uncertainty-Aware Planning Enhances Information Seeking in LLMs"
_NeurIPS.cc/2024/Conference — NeurIPS 2024 poster_

### Official Review · Reviewer_c9Ar · 2024-06-24

**Soundness:** 3
**Presentation:** 3
**Contribution:** 3
**Rating:** 7
**Confidence:** 3

**Summary:**

This paper presents the Uncertainty of Thoughts (UoT) algorithm designed to enhance large language models (LLMs) by enabling them to actively seek information through effective questioning. UoT incorporates three key components: an uncertainty-aware simulation approach to model possible future scenarios, uncertainty-based rewards motivated by information gain, and a reward propagation scheme to select optimal questions. The algorithm is evaluated across multiple LLMs in various scenarios, including medical diagnosis, troubleshooting, and the ‘20 Questions’ game. The results show a significant improvement in task completion rates and efficiency, demonstrating the effectiveness of UoT in reducing model uncertainty and enhancing information-seeking behaviour.

**Strengths:**

1. The introduction of uncertainty-aware planning and reward propagation for question generation is a novel and significant contribution to improving LLMs’ performance in interactive environments.
2. The experiments are well-designed, covering diverse scenarios (medical diagnosis, troubleshooting, 20 Questions) and multiple LLMs, providing robust evidence of UoT’s effectiveness.
3.  The algorithm achieves substantial improvements in success rates and efficiency and the use of entropy and information gain to measure and reduce uncertainty is effective.
4. The codes are available for the replication and future research.

**Weaknesses:**

1. The UoT framework, with its simulation and reward propagation components, might be complex to implement and integrate into existing LLM systems without significant computational resources. Also, this paper lacks of the analysis and comparison of inference time between each method, which will largely  affect the user experience.
2. The paper primarily uses success rates and conversation lengths as evaluation metrics, but additional qualitative analyses of the generated questions and their impact on the decision-making process could provide deeper insights. For instance, the impact of UoT on the user experience, particularly in terms of the naturalness and relevance of the generated questions, is not thoroughly explored.

**Questions:**

Given the complexity of UoT, do you think the inference delay of UoT will affect the user experience in real applications?

**Limitations:**

The limitations are discussed in Appendix H Limitation and Future Work.

---

> ### Author Rebuttal · Authors · 2024-08-07
>
> We highly value the feedback you provided. Your suggestions have prompted us to refine certain aspects of our work. We hope the following clarifications will satisfactorily address the points you raised.
>
> > Q1: The paper lacks analysis and comparison of inference times, which are crucial for user experience. Do you think UoT's inference delay will impact user experience in real applications?
>
> Sorry for the missing inference time. We also discuss the computational efficiency of our method in Table 2 of our paper by examining token consumption in each turn interaction. Based on the results, we can roughly estimate the inference time for different methods. Meanwhile, following your suggestion, we conducted this inference efficiency experiment. We provide the results and analysis below and plan to include these results in Section 3.3 Analysis in the next revision.
>
> We supplement inference efficiency using the Time to One Turn Conversation (TTOTC, measuring the time LLMs received an answerer's last round of responses to when the question is asked in the current round). We randomly sample 10 times from each dataset and conduct tests during both peak and off-peak hours of the GPT-4 API twice. The average TTOTC is reported below.
>
> Results Table
> | Dataset | Avg. Time (s) |
> | -------  | ------------- |
> | Common           | 5.14   |
> | Thing           | 9.77       |
> | DX               | 3.64       |
> | MedDG        | 9.29       |
> | FloDial          | 4.07       |
>
>
> ***Impact of delay***
>
> Inference delay is crucial in real-time applications. Long inference times can impact user experience. While UoT may introduce delays, its benefits in information acquisition and task success can offset this. For example, in medical diagnosis, accuracy is often more important than speed. For example, in medical diagnosis, accurate and effective information acquisition may be more important than quick but less effective responses.
>
> ***Ways to mitigate the influence of inference latency***
> * With the development of LLMs, inference efficiency will improve for both open-source and commercial models, also enhancing the efficiency of our UoT method.
>
> * One simple way to optimize the complexity of UoT is to prune our UoT search tree, which we also introduce in Section 2.7 Extensions and Discussion. As the results shown in Table 1 and Table2, the Pruned UoT can still be the SOTA method in 4/5 datasets and only require half token consumption, and It will double the inference efficiency according to our test.
>
> * In practice, we can train the LLM to be more efficient policy engines that can directly generate good questions without exhaustive tree search. For example, previous work uses tree search for a teacher model and distillation to train a student model to replicate the teacher's output without tree search [1]. Also, we can enhance LLMs through SFT and RLHF.
>     * RLHF with UoT: UoT helps generate preference data by evaluating and selecting optimal questions based on expected rewards, refining LLMs with nuanced data.
>     * SFT with UoT: UoT collects successful trajectories in contexts like 20 Questions, medical diagnosis, and troubleshooting, providing valuable training data for LLM refinement. These approaches can improve LLMs' autonomous information-seeking capabilities across various domains and make it more efficient in real applications.
>
> [1] [Learning by Distilling Context](https://arxiv.org/abs/2209.15189). Charlie Snell, Dan Klein, Ruiqi Zhong.
>
>
> > Q2: Additional qualitative analyses of the generated questions and their impact on decision-making are needed. The impact of UoT on user experience, especially regarding the naturalness and relevance of the questions, is not thoroughly explored.
>
> We acknowledge that the current evaluation metrics do not fully reflect the user experience in real applications.
>
> We conducted simple experiments based on the GPT-4 UoT method to test user satisfaction, generation fluency, and the relevance of generated questions using the MedicalDG dataset. Two annotators rated these factors on a scale from 0 to 5, resulting in average scores of 4.27 for user satisfaction, 5.00 for generation fluency, and 4.02 for the relevance of generated questions.
>
> We will include these experimental results in the final version and further design user experience studies to gather feedback on the naturalness and relevance of the generated questions. Combining this with the analysis of task success rates and conversation lengths, we will provide a more comprehensive evaluation and deeper insights into the decision-making process and user experience.

---

> > ### Author Response · Authors · 2024-08-08
> > **Supplementation for Inference Time Comparison**
> >
> > Dear reviwer c9Ar:
> >
> > We supplement our comparison of all the methods reported in the paper by utilizing GPT-4 and conducting experiments in a closed-set setting. We randomly sample 10 times from each dataset and perform tests during both peak and off-peak hours of the GPT-4 API, repeating the process twice. The average Time to One Turn Conversation (TTOTC, in seconds) is reported below.
> >
> > | Dataset | DP  | PP    | CoT  | Reflexion | Original-ToT | Ad.-ToT | UoT  |
> > | ------- | --- | ----- | ---- | ------ | ------------ | ------- | ---- |
> > | Common  | 1.00 | 15.24 | 4.99 | 14.88  | 10.39        | 6.83    | 5.14 |
> > | Thing   | 0.99 | 17.27 | 5.20 | 15.34  | 11.08        | 10.53   | 9.77 |
> > | DX      | 1.26 | 11.24 | 6.83 | 13.81  | 9.47         | 4.80    | 3.64 |
> > | MedDG   | 1.25 | 12.11 | 7.48 | 14.88  | 9.91         | 9.96    | 9.29 |
> > | FloDial | 1.53 | 15.54 | 7.17 | 15.73  | 10.62        | 6.12    | 4.07 |
> >
> > The results indicate that the inference efficiency of the UoT method surpasses that of the ToT, Reflexion, and PP methods. Additionally, UoT's inference time is comparable to that of the CoT method across all datasets.

---

> > ### Comment · Reviewer_c9Ar · 2024-08-12
> >
> > Thanks for the detailed response!

---

### Official Review · Reviewer_W6dt · 2024-07-10

**Soundness:** 3
**Presentation:** 4
**Contribution:** 3
**Rating:** 7
**Confidence:** 4

**Summary:**

The authors propose an uncertainty-aware information-seeking framework. This approach involves having the LLM simulate future scenarios and select the question that maximizes information gain. They evaluate their method using the latest LLMs across various benchmarks and introduce new datasets specifically designed to assess the LLMs’ information-seeking capabilities.

**Strengths:**

- The problem and the uncertainty-based approach are well-motivated, with relevant real-life examples.
- The idea is clearly explained, providing a sufficient amount of detail.
- The performance of their method surpasses that of the baselines, as demonstrated in Table 1.

**Weaknesses:**

- My biggest concern is the evaluation of the methods using GPT-4 as the answerer. In UoT, the LLM simulates future scenarios by itself and selects an optimal question. Since GPT-4 is the answerer in the evaluation rather than a human, this simulation might be more successful compared to real-life scenarios where a real person is the answerer. The authors should include at least one human-based experiment to validate their approach in real-life situations.
- The authors should discuss the cost of their approach through experimental analysis.
- I would like to see an evaluation of how robust this framework is to different prompts during the internal steps.

**Questions:**

Do you always assume $p(x_i)$ is uniformly distributed? Can't you use token probabilities to set these probabilities?

**Limitations:**

Authors didn't add a limitations section but I think, they can add their methods' cost as a limitation.

---

> ### Author Rebuttal · Authors · 2024-08-07
>
> We sincerely appreciate your detailed feedback and suggestions. Your comments have been very helpful in guiding us to enhance our work. Please find our clarifications below, which we hope will resolve your concerns.
>
> > Q1: The authors should include human-based experiments
>
> As you suggested, we run this human-based answerer experiment in 20 Questions (Common) and Medical Diagnosis (MedDG) based on GPT-4 (close-set), and it shows that the results(Success Rate) for both DP and UoT method are very close to the GPT-based answerer experimental results.
>
> Due to the time limitation, we only conduct the current experiments, and will further supplement the experiments in different LLMs, datasets and baselines to Section 3.2 Performance to the next revision.
>
> | Dataset    | DP   | UoT  |
> | ---------- | ---- | ---- |
> | Common     | 45.9 (50.5) | 72.1 (71.2) |
> | MedDG | 74.0 (72.3) | 86.0 (88.0)  |
>
> *PS: | Results of human-based experiments (GPT-based answerer experimental results) |
>
> > Q2: The authors should discuss the cost of their approach through experimental analysis.
>
> We discuss the computational efficiency of our method in Table 2 by examining token consumption in each turn interaction. Based on your suggestion, we carried out this experimental cost calculation, and we will incorporate these analysis into Section 3.3 Analysis in the final revision.
>
> According to our calculations, UoT with a depth of 3 consumes approximately 9.2k tokens per turn, while Pruned UoT with a depth of 3 uses about half that amount. Based on the official GPT-4-turbo pricing (USD 10.00 per 1M tokens) and the average conversation length for each dataset, the cost per task is USD 1.24 for Common, USD1.64 for Things, USD 0.19 for DX, USD1.08 for MedDG, and USD1.09 for FloDial. The total cost to run one experiment across these five datasets is USD 137.86, USD 3053.17, USD 20.09, USD 133.40, and USD166.10 respectively. If we use the latest GPT-4o API (gpt-4o-2024-08-06) at $2.5 per 1M tokens, the costs would be one-fourth of the current amounts. Especially, the cost is dominated by input tokens so the cost of output tokens is small enough to be neglected.
>
> > Q3: I would like to see an evaluation of how robust this framework is to different prompts during the internal steps.
>
> To validate the robustness of our method to different prompts, we rephrase the current prompt and conduct close-set experiments in 20 Questions (Common) and Medical Diagnosis (MedDG) based on GPT-4, using the original experimental settings.
>
> Original Prompt:
> ```
> Please design a question about X and can only be answered by YES or NO. {asked} Then classify the possible X above based on this question. If the answer is 'YES', put this X into 'YES: ...', otherwise to 'NO: ...'. Finally calculate how many X in YES and NO.
>
> Notably, this question should fulfill that the count of YES and NO are almost the same with a permissible discrepancy of no more than one!
>
> You should think about best {m} questions to respond to.
> ```
>
> Rephrased Prompt:
> ```
> Please formulate a question regarding X that can be answered solely with YES or NO. {asked} Then, categorize each possible X based on the response to this question. Place the Xs with a 'YES' answer under 'YES: ...' and those with a 'NO' answer under 'NO: ...'. Afterwards, tally the number of Xs in both the YES and NO groups.
>
> It is important that the question ensures the counts of YES and NO are almost equal, allowing for a difference of no more than one.
> Consider the most appropriate {m} questions to answer.
> ```
>
> Experiment results:
>
> | Dataset | SR          | MSC         | MCL         |
> | ------- | ----------- | ----------- | ----------- |
> | Common  | 70.3 (71.2) | 11.1 (10.8) | 13.7 (13.5) |
> | MedDG   | 86.0 (88.0) | 2.7 (2.6)   | 3.0 (2.9)   |
>
> *PS: | Result of experiment with rephrased prompts (original result in the paper) |
>
> As the results shown in the above table, the metrics (SR, MSC, and MCL) for the rephrased prompts are very close to those for the original prompts. For example, the SR in the Common dataset changed from 71.2 to 70.3. These minimal differences indicate that our method remains robust and consistent, regardless of prompt phrasing. We will further supplement the experiments in different LLMs, datasets and baselines in our final version.

---

> ### Author Response · Authors · 2024-08-07
> **Response to Question: Do you always assume p(x_i) is uniformly distributed? Can't you use token probabilities to set these probabilities?**
>
> Thanks for your suggestion. Our current implementation assumes equal probabilities for all possibilities for simplicity. However, the algorithm description in the paper in fact allows for non-uniform starting probabilities, assigning higher probabilities to more common diseases: in the example in lines 156-159, we assign non-uniform probabilities $P(x_1)=0.2, P(x_2)=0.3, P(x_3)=0.5$. These prior probabilities will naturally affect the computation of expected rewards in UoT, giving greater weight to more probable diseases.
> Additionally, we can further extend this approach to incorporate a probabilistic model between possibilities and answers: e.g., a patient with Covid-19 is more likely to have a fever, but not 100% likely. To do this, we can use a Bayesian approach.
>
> Consider a scenario with disease candidates $\Omega$ = {flu($H_1$), gastritis($H_2$), Covid-19($H_3$), appendicitis($H_4$)}. The response "Yes, I have a fever" may indicate different probabilities for each disease. Assume their prior probabilities are equal: $P(H_1) = P(H_2) = P(H_3) = P(H_4) = 0.25$ before receiving the response. Now, we pose the question, "Do you have a fever?" and receive an affirmative answer.
>
> We can utilize LLMs to estimate likelihood probabilities based on their knowledge. This can be done verbally, for example, by saying, "I think COVID-19 typically causes fever. The probability of having a fever in confirmed cases is approximately 40%." Alternatively, like your suggestion, we can prompt an LLM to answer a question while assuming a specific possibility $\omega$, Covid-19, and then extract its logits probability for the tokens 'Yes'.
>
> Then, we can obtain likelihood probabilities for these disease possibilities :
> * Flu $P(E|H_1) = 0.8$
> * Gastritis $P(E|H_2) = 0.2$
> * Covid-19 $P(E|H_3) = 0.4$
> * Appendicitis $P(E|H_4) = 0.4$.
> Where $E$ represents the observed evidence "the patient has a fever."
>
> The total probability $P(E) = \sum_{i} P(E|H_i)*P(H_i)$, which equals 0.45.
>
> According to Bayes' rule:
> \begin{equation}
> P(H|E) = P(E|H)*P(H)/P(E)
> \end{equation}
>
> We can calculate the posterior probability for each disease $P(H|E)$. In multi-turn conversations, we can use the posterior probability from the previous turn as the prior probability for the next turn. These posterior probabilities can be used to narrow down the set of possibilities, serving as prior knowledge for LLMs to generate questions, calculate the expected reward or make final decisions.

---

> > ### Comment · Reviewer_W6dt · 2024-08-08
> >
> > Thanks to the authors for their response and additional experiments. Their answer seems satisfying, and I will increase my score to 7.

---

> > > ### Author Response · Authors · 2024-08-08
> > > **Response to Reviewer W6dt**
> > >
> > > Thanks for your appreciation. Your insights and suggestions have been invaluable in enhancing the quality of our work. We will further modify our paper for the final version.

---

### Official Review · Reviewer_YbAg · 2024-07-11

**Soundness:** 4
**Presentation:** 4
**Contribution:** 4
**Rating:** 10
**Confidence:** 2

**Summary:**

The paper addresses the problem of how to guide an LLM to find the right answer to a given question, in cases additional information must be elicited from the agent (possibly human) asking the question before knowing the right answer. The proposed algorithm, called Uncertainty of Thoughts (UoT), works by simulating possible questions to ask and answering it might receive to choose the question that will minimize the uncertainty over the right answer. UoT spans a tree of questions and answers, and chooses which question to ask at the end by estimating the value of information gained by the different questions. The authors perform a very extensive experimental evaluation of UoT, on 3 benchmarks, 5 LLMs, and quite a few LLM question-answering frameworks. The results are very impressive, showing that UoT can find the right answer faster and in more cases than all other options in almost all cases. Moreover, they created benchmarks for this important problem for the community to use.

**Strengths:**

In general, I am very excited about this work due to the following reasons:
-	The problem they identified and formalized is very important and I believe relatively understudied in a formal way in the world of LLMs
-	The proposed algorithm – UoT – is novel and exciting. It is likely to open many directions for future work
-	The evaluation is very extensive and convincing.
-	The appendices seem to cover most questions and justify more design choices (at least empirically).

**Weaknesses:**

Some design choices are quite ad-hoc, in particular in how the reward function is defined and propagated. That being said, the authors support most choices empirically in the appendix.

Minor comment:
-	In page 5: “… over the typical ranges we encounter.” – do you mean typical ranges of IG values?

**Questions:**

1.	For the case where the answers are open ended, I do not fully understand how would you branch over the possible answers and relate them to limiting the set of possible answers. It seems to me that you would need some way to relate every open ended answers to the set of possible “right answers” that are still consistent. Is this what is intended?

2.	I don’t fully understand why do we need the accumulated reward function. The basic reward function measures information gain for the leaf nodes. The expected reward function aggregates recursively the rewards from leaf nodes. Why do we need the leaf node to accumulate the rewards gained up to it? It would be more natural to have the reward at the leaf measure the information gained by reaching that leaf node, and then have the expected reward aggregate these values as it is defined now (without the accumulated reward).

3.	Currently the tree is limited to a fixed depth. Why not use an MCTS type of search where the depth is not fixed (e.g., UCT)?

**Limitations:**

No limitations due to negative societal impact I can think of.

---

> ### Author Rebuttal · Authors · 2024-08-07
>
> Thank you very much for recognizing the value of our work and providing valuable suggestions. Your feedback is instrumental in enhancing the quality of our research. Below are some clarifications that we hope will address your concerns.
>
>
> > Q1: For the case where the answers are open ended, I do not fully understand how would you branch over the possible answers and relate them to limiting the set of possible answers. It seems to me that you would need some way to relate every open ended answers to the set of possible “right answers” that are still consistent. Is this what is intended?
>
> A somewhat complex aspect of our algorithm lies in how the Answer Simulation step works, as there is an apparent conflict between the Answerer providing open-ended answers, and UoT grouping responses into 2 categories (affirmative and negative) at each Answerer node.
>
> To resolve this apparent conflict, we need to clarify that "real" answers (those given by the user in the actual conversation) are open-ended, as there are no restrictions on how the Answerer (e.g. the human) can respond. However, for the "imagined" answers in UoT's simulated futures, we consider these to have only 2 categories (affirmative and negative), as this is necessary to compute meaningful uncertainty metrics. To do so, at the Answerer nodes, instead of using an LLM to generate answers, we instead prompt the LLM to decide which of the possibilities among the current possibility set would lead to an affirmative answer, and which would lead to a negative answer. In this way, we partition the current possibility set into 2 subsets, which are then used as the current possibility sets of the 2 children of the current node. (Section 2.3, lines 124-134)
>
> > Q2: Why an accumulated reward function is necessary
>
> Accumulated reward is essential for considering long-term effects in dynamic environments, rather than just immediate effects. By summing the rewards of a node $v$ and all its ancestor nodes, we reflect the effectiveness of past decisions, evaluating the total reward along the entire conversation path leading to a leaf node. The cumulative reward at a leaf node represents the total reward at the end of the conversation.
>
> For instance, in a 3-step conversation, the immediate reward function $R_u(v)$ captures the information gained from receiving the answer at node $v$. However, the total information gained by the end of the conversation is the sum of the information acquired over the three rounds. Therefore, we accumulate the immediate rewards  $R_u(v)$ of a leaf node and its parent nodes to compute the accumulated reward, thereby accounting for the information gained across multiple rounds of the conversation.
>
>
> > Q3:  Currently the tree is limited to a fixed depth. Why not use an MCTS type of search where the depth is not fixed (e.g., UCT)?
>
> Thanks for your suggestion. Our current approach only considers fixed depth for simplicity.  We can use our current information gained based reward design with variable depth to extend dynamically based on the promising outcomes of the simulations.

---

> > ### Comment · Reviewer_YbAg · 2024-08-13
> > **Still excited about this paper, even though I disagree with some points**
> >
> > Re. your answers to Q1 and Q3, thanks! they helped clarify these points. I think it is worthwhile for you to revisit the text in the paper to clarify there too.
> > Re. your answer to Q2, I disagree with "Accumulated reward is essential for considering long-term effects in dynamic environments." For example, see all the success of MCTS algorithms in games, where there is no accumulated reward, only the value at the game tree leaves (win/lose). In particular in your work, since the objective is to plan which sequence of questions to ask in order to gain the most information, why do intermediate gains matter? If two sequences of questions eventually obtain the same information from the user, does it matter if one sequence obtained some information earlier than the other? If the intermediate rewards were perfect, then it does not matter, but since they are only estimates of the future reward, why use them and not the eventual information at the leaf?

---

> > > ### Author Response · Authors · 2024-08-14
> > > **Discussion about the meaning of accumulated reward and intermediate gains**
> > >
> > > Thank you for your valuable feedback. We'd like to clarify the meaning of Accumulated Reward and Intermediate Gains in our work.
> > >
> > > The main reason why we need to accumulate rewards is that our immediate rewards $R_u(v)$ actually measure the change in uncertainty (i.e., entropy) when we receive the answer at that node (Lines 180-182), rather than the amount of entropy at the intermediate node. If we consider the analogy to games such as chess, here our immediate rewards are analogous to the change in position evaluation when making a particular move. As you mentioned, in MCTS, decisions are generally made based on the final position evaluation (or value) at a leaf, which is essentially the sum of these changes over the whole game. Thus, our accumulated reward is analogous to the final position evaluation (or value) in games like chess. Indeed, implementing our rewards by evaluating the entropy at each leaf node as value would also have been a reasonable (and more or less equivalent) way to design our rewards. Additionally, in MCTS, focusing solely on the final node's reward may introduce bias, especially with shallow or narrow simulations that overlook intermediate values. Enhanced MCTS algorithms use trajectory-based rewards to better estimate future gains. Our approach, Accumulated Reward, sums immediate rewards from the start to a specific node, akin to value estimation in reinforcement learning. This enables our model to prioritize paths with higher overall information gain, even under constrained simulations. We aim to refine our terminology in future versions to better align with value estimation concepts.
> > >
> > > When designing our reward scheme, we also took inspiration from works like MuZero [1] by DeepMind, which accumulates rewards along a trajectory during MCTS to update the Q-values for each state-action pair (Equations 3 and 4 in their work). Besides, recent research in LLM planning, such as Math-Shepherd [2], considers the quality of intermediate steps (Equation 4 in paper), and RAP [3] (also based on MCTS) selects paths based on Q-values derived from the average of future rewards over multiple steps (Equation 2 of their paper). Inspired by these approaches, we also considered incorporating Intermediate Gains into our Accumulated Reward to enhance the impact of different steps within the simulation.
> > >
> > > We also plan to further discuss the motivation behind our reward design in the final version of our paper, specifically in Section 2.5 (Question Selection Via Reward Propagation) and the appendix. We will include experimental results based on your suggestion (eventual information at the leaf node as a basis for path selection) and provide additional discussion and comparison.
> > >
> > > [1] Mastering Atari, Go, Chess and Shogi by Planning with a Learned Model.
> > >
> > > [2] Math-Shepherd: Verify and Reinforce LLMs Step-by-step without Human Annotations.
> > >
> > > [3] Reasoning with Language Model is Planning with World Model.

---

### Official Review · Reviewer_moAn · 2024-07-12

**Soundness:** 3
**Presentation:** 3
**Contribution:** 3
**Rating:** 6
**Confidence:** 3

**Summary:**

The paper introduces Uncertainty of Thoughts (UoT), aimed at enhancing the ability of large language models (LLMs) to actively seek information by asking effective questions. UoT integrates an uncertainty-aware simulation method, uncertainty-based rewards motivated by information gain, and a reward propagation scheme to optimize question selection. The primary innovation lies in the model’s capability to project possible future scenarios and choose questions that maximize expected rewards, thereby efficiently gathering necessary information. This methodology has been shown to improve the performance of multiple LLMs in tasks such as medical diagnosis, troubleshooting, and playing the ‘20 Questions’ game.

**Strengths:**

1. The idea of combining uncertainty-aware simulations with a reward-based system to guide LLMs in asking more effective questions is novel and interesting.
2. The authors provide comprehensive experiments to validate their approach, comparing it against baseline models on three different tasks, which shows improvements in effectiveness.
3. The paper is well-written and the problem formulation is mostly easy to follow.

**Weaknesses:**

1. Questionable Baseline Performance: There is a notable performance discrepancy between the baseline models (CoT and ToT) and direct prompting, especially in the 20 Questions task evaluated by GPT-4. Direct prompting achieves a 48.6% success rate in the open set, while CoT drops significantly to 13.5%. In the closed set, while direct prompting achieves 50.5%, CoT and ToT only reach 20.7% and 28.8% respectively. Such substantial decreases in performance warrant further investigation.

2. Lack of Explanation: The paper lacks a detailed explanation of the prompts and setups used for CoT and ToT. Understanding why these methods underperform could be clarified through additional analysis and qualitative results.

3. In the checklist, the author claimed yes to having the experimental statistical significance. But I did not see this in any table.

4. Q: Did you consider the LLM’s own uncertainty? For example, the LLM proposed two questions but it could be more certain about one of the questions. In the paper, it seems that you considered each question to be equally possible when sampling.

5. It is unclear to me how information gain is assessed in scenarios with open-ended responses. Did you still use the LLM to generate the output with the fixed template at the end so that you can apply the same evaluation framework as the closed set setting? Otherwise, how to know the reward?

**Questions:**

1. Could you provide some analysis of why there is a notable performance decrease between the baseline models (CoT and ToT) and direct prompting?
2. How is information gain computed in the open-ended answers setting?

**Limitations:**

The limitations are described.

---

> ### Author Rebuttal · Authors · 2024-08-07
>
> We are grateful for your thorough review and insightful comments. Your feedback has encouraged us to refine and improve our work. We hope the clarifications below will adequately address the issues you mentioned.
>
> > Q1:  There's a significant performance gap between baseline models (CoT and ToT) and direct prompting in the 20 Questions task evaluated by GPT-4. Direct prompting achieves much higher success rates. The paper doesn't detail the prompts and setups for CoT and ToT, requiring further investigation to understand their underperformance.
>
> Sorry for the missing of CoT and ToT's prompt in our paper. We use the standard idea of CoT and ToT (self reward). We provide the prompts here in a 20 question scenario here and will supplement the comprehensive prompts for all baseline methods in our final version.
>
> | CoT Prompts in 20 Questions |
> |---------------------------------------------------|
> | *Prologue*  |
> | Let us play the game of 20 questions. I am impersonating the thing, X. You will ask me up to 20 questions that start with ’Is X’ and can only be answered by yes or no, and I will answer each one truthfully based on being X. Let us begin. Ask me the first question. Reply to me with your first question. |
> | *Prompt for Generating Question and Explanation*|
> | What’s your next question? Let’s think step-by-step and reply me with your explanation.<br>Your answer should be:<br>Explanation: [insert step-by-step analysis here]<br>Question: [next question] |
> | *Additional Reminder in Q14 - Q20* |
> | Note that you should guess and ask what X exactly is from now on. X is possible a: [item_list_str], or other. The question must start with ’Is X ...  |
>
> | ToT Prompt for 20 Question game. [C1] is the count of questions asked. The ‘information gained’ marked blue represents the previous interaction history.|
> |---------------------------------------------------|
> | *Prologue*  |
> | You are playing the game of 20 questions. I am impersonating the thing, X. You will ask me up to 20 questions that start with ’Is X’ and can only be answered by yes or no, and I will answer each one truthfully based on being X. <br>[C1] questions have been asked. And now we know: <br>[information gained] <br>Design a question about X and can only be answered by YES or NO.                   |
> | *Additional Reminder in Q14 - Q20*  |
> | Note that you should guess and ask what X exactly is from now on. X is possible a: [item_list_str]. The question must start with ’Is X ...  |
>
> **Analysis of a notable performance decrease in CoT and ToT compared to Direct Prompting (DP)**
>
> **CoT**: CoT method focuses more on analyzing possible answers based on current responses and interactions rather than actively seeking information to further reduce uncertainty. CoT often performs extensive self-analysis after obtaining some initial information in the early rounds. It then tends to ask questions related to what it believes might be the correct answer. However, asking specific questions without sufficient information can easily lead to failure. This is especially true in tasks like 20 Questions, where there are many interaction rounds.
> ```
> Example:
> True Option: Flamingo
> [ROUND 8]
> Questioner:
> Explanation: So far, we have established that X is a living thing, an animal, a vertebrate, a bird, a large bird, lives near water, and eats fish.  This information narrows down the possibilities significantly.  A large bird that lives near water and eats fish could be a pelican, a seagull, a heron, or an eagle, among others.  To further narrow down the possibilities, we can ask about a specific characteristic that only some of these birds have.  For example, not all of these birds can dive underwater to catch their prey.
> Question: Can X dive underwater to catch its prey?
> Answerer:
> No, X does not dive underwater to catch its prey.
> [ROUND 9]
> Questioner:
>
> Explanation: So far, we know that X is a living thing, an animal, a vertebrate, a bird, a large bird, lives near water, eats fish, but does not dive underwater to catch its prey.  This information narrows down the possibilities significantly.  A large bird that lives near water and eats fish but does not dive underwater could be a type of heron, stork, or pelican.  To narrow it down further, we can ask about a specific characteristic of one of these birds.  For example, pelicans are known for their large throat pouch used for catching prey and draining water before they swallow.
> Question: Does X have a large throat pouch?
> Answerer:
> No, X does not have a large throat pouch.
> ```
>
> **ToT**: In the original ToT, the reward function is based on self-reward. Self-reward lacks the principle gradually guiding LLMs to choose questions that effectively narrow down the possibilities set, as seen in UoT. When relying on self-reward, the model often selects more general questions, making it difficult to obtain information about the targeted possibility.
> ```
> Example:
> True Option: Meerkat
> [ROUND 1]
> Questioner: Is X a man-made object?
> Answerer: No, X is not a man-made object.
> [ROUND 2]
> Questioner: Is X a food?
> Answerer: No, X is not food.
> [ROUND 3]
> Questioner: Is X a living organism?
> Answerer: Yes, X is a living organism.
> [ROUND 4]
> Questioner: Is X a plant?
> Answerer: No, X is not a plant.
> [ROUND 5]
> Questioner: Is X a mammal?
> Answerer: Yes, X is a mammal.
> [ROUND 6]
> Questioner: Is X a domesticated mammal?
> Answerer: No, X is not a domesticated mammal.
> [ROUND 7]
> Questioner: Is X a carnivorous mammal?
> Answerer: No, X is not a carnivorous mammal.
> ```
> **DP**: The DP method poses questions that are more coherent and logical overall. This is because the DP method uses shorter text inputs in LLMs, allowing for a better understanding of the conversation history. It avoids being misled by premature judgments in multiple rounds of interaction (such as CoT) since it analyzes implicitly, without relying on explicit but insufficient self-reward. This enables it to adjust the questioning logic in a timely manner.

---

> ### Author Response · Authors · 2024-08-07
> **Response to Remaining Questions**
>
> > Q2:  In the checklist, the author claimed yes to having the experimental statistical significance. But I did not see this in any table.
>
> Sorry for the missing significance test results in main body of our paper. Previously, we conducted three experiments on five datasets using Llama 3 and GPT-4 to compare the performance of Direct Prompting (DP) and UoT methods in a closed-set setting for significance test. Due to the LLM API quota limitation, our number of experiments was restricted. To determine whether the differences in success rates (SR) between the two methods were statistically significant, we performed a t-test. The results are as follows:
>
> GPT-4
> | Dataset     | DP | UoT | t-Statistic | p-Value | Significance Conclusion |
> |-------------   |-----------------|--------------|-------------|---------|-------------------------|
> | Common   | 49.0   | 70.9 | -10.8 | 0.00041   | Significant (p < 0.05)  |
> | Thing         | 30.8   | 36.8 | -8.04  | 0.00129   | Significant (p < 0.05) |
> | DX             | 89.4   | 97.0 | -3.11 | 0.03581   | Significant (p < 0.05) |
> | MedDG      | 74.9  | 87.9 | -7.33 | 0.00185   | Significant (p < 0.05) |
> | FloDial       | 42.5  | 67.8 | -19.8  | 0.00004  | Significant (p < 0.05)  |
>
> Llama 3
>
> | Dataset     | DP | UoT | t-Statistic | p-Value | Significance Conclusion |
> |-------------   |-----------------|--------------|-------------|---------|-------------------------|
> | Common   | 47.7   | 56.5         | -4.39        | 0.01180   | Significant (p < 0.05)  |
> | Thing         | 14.8            | 24.8   | -16.0        | 0.00009   | Significant (p < 0.05) |
> | DX             | 80.1            | 90.1   | -4.65        | 0.00966   | Significant (p < 0.05)  |
> | MedDG      | 61.3            | 64.6         | -4.15        | 0.01426   | Significant (p < 0.05) |
> | FloDial       | 29.9            | 46.4         | -10.5        | 0.00047  | Significant (p < 0.05)  |
>
> The t-test results indicate that UoT significantly outperform DP five datasets (p < 0.05), as evidenced by their higher mean scores. We will supplement the significance test for remaining LLMs, methods comparison and settings in the final version and add the results in Section 3.2 Performance.
>
> > Q3: Did you consider the LLM’s own uncertainty? For example, the LLM proposed two questions but it could be more certain about one of the questions. In the paper, it seems that you considered each question to be equally possible when sampling.
>
> We consider each question to be equally likely for the following reasons:
>
> * Simplified Calculation: In the absence of prior knowledge, treating each question as equally likely simplifies the computation process. This approach ensures that the model, when lacking sufficient information, does not favor any specific question, making the calculations more straightforward and unbiased.
>
> * Fairness: At the early stages, considering each question as equally likely ensures that all potential questions are treated fairly. This is crucial for an exploration strategy, as it prevents premature convergence on specific questions, thereby more comprehensively covering the possible answer space.
>
> We can also improve the model's performance by leveraging the inherent uncertainty of the LLM. For example, we can use below confidence-based sampling method:
>
> * Confidence-Based Sampling: The LLM can provide confidence scores for each generated question. We can then perform weighted sampling based on these scores instead of equal probability sampling. For example, if the LLM is more confident about a particular question, it can be given a higher weight, making it more likely to be selected.
>
> > Q4: How is information gain assessed and computed in scenarios with open-ended responses, and did you use a fixed template for LLM-generated outputs to apply the same evaluation framework as the closed set setting?
>
> A somewhat complex aspect of our algorithm lies in how the Answer Simulation step works, as there is an apparent conflict between the Answerer providing open-ended answers, and UoT grouping responses into 2 categories (affirmative and negative) at each Answerer node.
>
> To resolve this apparent conflict, we need to clarify that "real" answers (those given by the user in the actual conversation) are open-ended, as there are no restrictions on how the Answerer (e.g. the human) can respond. However, for the "imagined" answers in UoT's simulated futures, we consider these to have only 2 categories (affirmative and negative), as this is necessary to compute meaningful uncertainty metrics. To do so, at the Answerer nodes, instead of using an LLM to generate answers, we instead prompt the LLM to decide which of the possibilities among the current possibility set would lead to an affirmative answer, and which would lead to a negative answer. In this way, we partition the current possibility set into 2 subsets, which are then used as the current possibility sets of the 2 children of the current node. (Section 2.3, lines 124-134)

---

> > ### Comment · Reviewer_moAn · 2024-08-11
> >
> > Thank you for conducting additional experiments and analysis. This addresses most of my concerns, and I'm happy to raise my score to a 6.

---

> > > ### Author Response · Authors · 2024-08-12
> > > **Response to Reviewer moAn**
> > >
> > > Thank you for your kind words. Your insights and suggestions have been instrumental in improving the quality of our work. We will make additional revisions to our paper for the final version.

---

> ### Author Response · Authors · 2024-08-11
> **Thank You for Your Review: We Welcome Any Further Questions or Concerns**
>
> Dear Reviewer moAn
>
> We sincerely appreciate the feedback you have provided. If there are any additional concerns or questions you may have, please do not hesitate to let us know. We are more than willing to discuss any further suggestions or issues you might have.

---

### Official Review · Reviewer_Ho3e · 2024-07-15

**Soundness:** 2
**Presentation:** 2
**Contribution:** 3
**Rating:** 6
**Confidence:** 3

**Summary:**

This paper presents an approach known as “uncertainty of thoughts (UoT)”, which builds upon related ideas of creating a tree of responses to answer a question, such as the “tree of thoughts” and related approaches. The key components of the system include 1) an approach to ask different types of questions and simulate yes/no sorts of answers, 2) uncertainty-based rewards that are somewhat motivated by information gain as measured by change in expected entropy, and 3) a reward propagation approach that computes sum of expected rewards. These combine to result in an approach where the system chooses a question from a set of questions until termination (which I did not fully understand). Experiments are conducted on benchmarks such as 20 Questions and Medical Diagnosis, and results generally seem favorable as compared to some baselines.

**Strengths:**

As per my understanding, the paper presents some novel ideas around using information gain to sequence questions using LLMs. I consider this a strength, but I did not see any description of how and when non-uniform probabilities are obtained for any set of remaining possibilities in the possibility space. The results also appear to be positive, from what I could understand.

**Weaknesses:**

I could not fully understand some important details in the paper. For instance, the open set case seems important but was not covered in sufficient detail in the paper.

I found many choices to be somewhat ad-hoc and without any clear basis. For instance, the choice of reward (after some sort of normalization) and the accumulation of rewards (that are all between 0 and 1) did not make sense to me. I think some choices make the work less principled, which affects the soundness of the work in my view.

Further details about my concerns are provided later in the section on Questions.

I am open to adjusting my score during the discussion period.

**Questions:**

Some questions and comments follow:

Is there ever a situation where the probabilities for each remaining possibility are non-uniform? If so, please explain how these probabilities are obtained. From what I understood, only the possibility set is reduced over iterations.

Is the output of the LLM in equation (1) obtained from a single inference query or multiple queries? If a single inference query, how does one ensure a variety of questions?

The output of the LLM in equation (2) seems problematic for open set case. Can you please explain how this is handled for open set?

Why does the modified reward in equation (10) work better empirically?

For the expected reward, why are downstream questions taken to be equally likely? I thought questions were decisions. If so, why is the max operator not used on those? Also, what is the meaning of the sum of the rewards in this situation? Why is this a suitable measure of reward? I believe the normalization nullifies the fact that each reward was originally a measure of information gain.

**Limitations:**

Much more discussion about limitations is needed.

---

> ### Author Rebuttal · Authors · 2024-08-07
>
> We greatly appreciate your insightful feedback. Below, we provide clarifications to address the concerns, which we will incorporate in next version.
>
> > Q1: The open set case seems important but was not covered in sufficient detail in the paper.
>
> Sorry for any confusion. We explain the open set setup in sections 3.1 and Appendix I.4. Here's a detailed explanation.
>
> In the closed set setting, our algorithm starts with a known possibility space $\Omega$. However, in the open set setting, we do not assume such a known $\Omega$. Instead, we generate $\Omega$ at the start of each conversation round (before UoT generates its questions), by prompting an LLM to generate a set of possibilities which are consistent with the current conversation history. Having generated $\Omega$, the rest of UoT proceeds exactly the same as in the closed set case to generate questions in each conversation round.
>
> Specifically, in the medical diagnosis and troubleshooting cases, since we have the initial symptom and issue descriptions available at the start, we use these to generate an initial possibility space. In each subsequent round, UoT refines the possibility space based on the current conversation history. In 20 Questions, since such initial descriptions are not available, we instead use Direct Prompting in Open-Set for the first three rounds. Afterward, UoT refines the possibility set each round.
>
> For the datasets Common, Things, DX, MedDG, and FloDial, we configure the size of the possibility set for each update round, setting them at 10, 10, 5, 5, and 5, respectively. This approach helps prevent the cognitive load increase and efficiency decrease associated with larger sizes while avoiding the limitations of focusing on too few items with smaller sizes. We experimented with sizes between 5 and 50 based on this rationale, and the final selection of these hyperparameters is guided by empirical performance evaluations.
>
> >  Q2: Some choices seem ad-hoc, like normalized rewards. Why not use the max operator? What does the sum of rewards represent? Why does the modified reward in equation (10) work better?
>
> **Meaning of Accumulated Reward**
>
> Accumulated reward is essential for considering long-term effects in dynamic environments, rather than just immediate effects. By summing the rewards of a node $v$ and all its ancestor nodes, we reflect the effectiveness of past decisions, evaluating the total reward along the entire conversation path leading to a leaf node. The cumulative reward at a leaf node represents the total reward at the end of the conversation.
>
> For instance, in a 3-step conversation, the immediate reward function $R_u(v)$ captures the information gained from receiving the answer at node $v$. However, the total information gained by the end of the conversation is the sum of the information acquired over the three rounds. Therefore, we accumulate the immediate rewards  $R_u(v)$ of a leaf node and its parent nodes to compute the accumulated reward, thereby accounting for the information gained across multiple rounds of the conversation.
>
> **Necessity of Normalization**
>
> Normalization ensures that reward values remain within a reasonable range during calculations, preventing extremely large or small values from skewing results. While normalization might affect the measurement of the original information gain, it ensures comparability of different rewards and stabilizes cumulative calculations. It's true that normalization can affect the measurement of original rewards (like information gain). However, the goal of normalization is to make rewards from different ranges comparable and to prevent certain rewards from disproportionately affecting the total. Choosing an appropriate normalization factor ensures that different types of questions or answers have reasonable weights in the calculations.
>
> **Why Downstream Problems are Considered Equally Likely**
>
> Downstream problems are considered equally likely to simplify calculations and ensure fair treatment of all potential questions in the absence of prior knowledge. This assumption allows the model to remain unbiased towards any specific question during calculations. In practice, other choices such as a "damping coefficient" which exponentially down weights each later round could also be used, based on our prior or application-specific knowledge.
>
> **Why Not Use the Max Operator**
>
> Using the max operator is also a reasonable approach, which assigns rewards based on the most favorable downstream outcome. Our approach of using the expected value instead can be considered as a more "risk-averse" approach which considers all downstream outcomes as possible scenarios. This risk-averse approach is justified due to the high unpredictability and difficulty in predicting conversation outcomes.
>
> **Meaning of the sum of the rewards**
>
> As discussed in “Meaning of Accumulated Reward,” the UoT uses the sum of rewards to guide question selection, balancing immediate and long-term benefits. Accumulated rewards indicate total uncertainty reduction, while expected rewards consider future gains, ensuring decisions that enhance task performance by reducing uncertainty across interactions.
>
> **Why does the modified reward in equation (10) work better empirically**
>
> Our modified reward design in equation (10), particularly setting lambda > 0, is intended as a straightforward method to incorporate our preference for a sharper reward, as it accelerates the decay of rewards as we move away from 0.5. Furthermore, it is also intended to penalize questions that are too specific when the set of possibilities remains relatively large as $|p_v^A - p_v^N|$ will be large. Meanwhile, there are alternative approaches to achieving this such as Logarithmic Transformation Scaling, Sigmoid Transformation Scaling and Piecewise Function Scaling. We provide the comparison results in Appendix B and the results show that our current setting provides better results, leading to our decision to use it.

---

> ### Author Response · Authors · 2024-08-07
> **Response to Remaining Questions**
>
> > Q3: Is there ever a situation where the probabilities for each remaining possibility are non-uniform? If so, please explain how these probabilities are obtained. From what I understood, only the possibility set is reduced over iterations.
>
> Our current implementation assumes equal probabilities for all possibilities for simplicity. However, the algorithm description in the paper in fact allows for non-uniform starting probabilities, assigning higher probabilities to more common diseases: in the example in lines 156-159, we assign non-uniform probabilities $P(x_1)=0.2, P(x_2)=0.3, P(x_3)=0.5$. These prior probabilities will naturally affect the computation of expected rewards in UoT, giving greater weight to more probable diseases.
>
> Additionally, we can further extend this approach to incorporate a probabilistic model between possibilities and answers: e.g., a patient with Covid-19 is more likely to have a fever, but not 100% likely. To do this, we can use a Bayesian approach.
>
> Consider a scenario with disease candidates $\Omega$ = {flu($H_1$), gastritis($H_2$), Covid-19($H_3$), appendicitis($H_4$)}. The response "Yes, I have a fever" may indicate different probabilities for each disease. Assume their prior probabilities are equal: $P(H_1) = P(H_2) = P(H_3) = P(H_4) = 0.25$ before receiving the response. Now, we pose the question, "Do you have a fever?" and receive an affirmative answer.
>
> We can utilize LLMs to estimate likelihood probabilities based on their knowledge. This can be done verbally, for example, by saying, "I think COVID-19 typically causes fever. The probability of having a fever in confirmed cases is approximately 40%." Alternatively, we can prompt an LLM to answer a question while assuming a specific possibility $\omega$, Covid-19, and then extract its logits for the tokens 'Yes'.
>
> Then, we can obtain likelihood probabilities for these disease possibilities :
> * Flu $P(E|H_1) = 0.8$
> * Gastritis $P(E|H_2) = 0.2$
> * Covid-19 $P(E|H_3) = 0.4$
> * Appendicitis $P(E|H_4) = 0.4$.
>
> Where $E$ represents the observed evidence "the patient has a fever."
>
> The total probability $P(E) = \sum_{i} P(E|H_i)*P(H_i)$, which equals 0.45.
>
> According to Bayes' rule:
> \begin{equation}
> P(H|E) = P(E|H)*P(H)/P(E)
> \end{equation}
>
> We can calculate the posterior probability for each disease $P(H|E)$. In multi-turn conversations, we can use the posterior probability from the previous turn as the prior probability for the next turn. These posterior probabilities can be used to narrow down the set of possibilities, serving as prior knowledge for LLMs to generate questions, calculate the expected reward or make final decisions.
>
> > Q4: Is the output of the LLM in equation (1) obtained from a single inference query or multiple queries? If a single inference query, how does one ensure a variety of questions?
>
> To clarify, we use a single inference query(one type prompt) to generate  $m$ questions. The complete prompt we use here is provided:
>
> ```
> Please design a question about X and can only be answered by YES or NO. {asked} Then classify the possible X above based on this question. If the answer is 'YES', put this X into 'YES: ...', otherwise to 'NO: ...'. Finally calculate how many X in YES and NO.
>
> Notably, this question should fulfill that the count of YES and NO are almost the same with a permissible discrepancy of no more than one!
>
> **You should think about best {m} questions to respond to. And your answer should be:**
>
> Question 1: Is X ...?
>
> YES: item1, item2, ...
>
> Count of YES: ...
>
> NO: item1, item2, ...
>
> Count of NO: ...
> ```
>
> > Q5: The output of the LLM in equation (2) seems problematic for open set case. Can you please explain how this is handled for open set?
>
> As mentioned earlier, we maintain a specific set of possibilities allowing LLMs to generate candidate questions and reset this set for the next turn based on the latest interaction. Therefore, $LLM_{ans}$ can still determine the further subsets $\Omega_v^A$ and $\Omega_v^N$ as it does in a closed set setting.

---

> ### Author Response · Authors · 2024-08-11
> **Thank You for Your Review: We Welcome Any Further Questions or Concerns**
>
> Dear Reviewer Ho3e,
>
> We sincerely appreciate your feedback and would like to inquire if there are any remaining concerns or questions that we can address. We are happy to communicate and address any of your suggestions or remaining concerns.

---

> ### Author Response · Authors · 2024-08-13
> **Gentle Follow-up: We Are Here to Address Any Further Questions**
>
> Dear Reviewer Ho3e,
>
> I hope this message finds you well. We want to express our gratitude once again for your valuable feedback on our submission. As the deadline is approaching, we kindly want to check in to see if there are any additional concerns or questions that you would like us to address.
>
> We are more than happy to further clarify or revise any aspects of our submission based on your input.
>
> Thank you so much for your time and consideration. We truly appreciate your efforts.
>
> Best regards,
>
> Authors

---

> > ### Comment · Reviewer_Ho3e · 2024-08-13
> > **Thanks**
> >
> > Thanks for your detailed responses. I'll take them into account during further discussions with other reviewers.

---

> > > ### Author Response · Authors · 2024-08-14
> > > **Sincere Thanks for Your Feedback**
> > >
> > > Thank you for your feedback and continued engagement with our work! We appreciate your time and consideration.

---

### Author Rebuttal · Authors · 2024-08-07

We sincerely thank all the reviewers for their helpful comments and suggestions. Here is a summary of our responses to address the major concerns of reviewers.

**1. Elaboration of Open Set Setting (Proposed by reviewer1[Ho3e15])**

In the closed set setting, our algorithm starts with a known possibility space $\Omega$. In the open set setting, we generate $\Omega$ at the start of each conversation round by prompting an LLM with the current conversation history. After generating $\Omega$, UoT proceeds as in the closed set case to generate questions.

**2. Clarification on Answer Simulation Step, and How Open-Ended Responses are Mapped to Categories (Proposed by reviewer2 moAn, reviewer3 YbAg)**

Our Answer Simulation step seems complex due to the apparent conflict between the Answerer providing open-ended answers, and UoT grouping responses into 2 categories (affirmative and negative) at each Answerer node. Below, we provide additional clarification to resolve this conflict, and will incorporate this into the explanations in the paper.

"Real" answers (those given by the user in the actual conversation) are open-ended, as there are no restrictions on how the Answerer (e.g. the human) can respond. However, for the "imagined" answers in UoT's simulated futures, we consider these to have only 2 categories (affirmative and negative), as this is necessary to compute meaningful uncertainty metrics. Thus, at the Answerer nodes, instead of using an LLM to generate answers, we instead prompt the LLM to decide which of the possibilities among the current possibility set would lead to an affirmative answer, and which would lead to a negative answer. In this way, we partition the current possibility set into 2 subsets, which are then used as the current possibility sets of the 2 children of the current node. (Section 2.3, lines 124-134)

**3 Approach to Deal with Non-Uniform Probabilities in Possibility Set (Proposed by reviewer1Ho3e15 and reviewer4 W6dt)**

Our current implementation assumes equal probabilities for all possibilities for simplicity. However, the algorithm description in the paper in fact allows for non-uniform starting probabilities, assigning higher probabilities to more common diseases: in the example in lines 156-159, we assign non-uniform probabilities $P(x_1)=0.2, P(x_2)=0.3, P(x_3)=0.5$. These prior probabilities will naturally affect the computation of expected rewards in UoT, giving greater weight to more probable diseases.

This framework can be further extended using a probabilistic model between possibilities and answers, by using a Bayesian approach. Specifically, we start by assigning non-uniform prior probabilities based on disease frequencies. At each Answerer node, for each possibility $\omega$, we estimate the likelihood of observing a 'Yes' or 'No' answer to the current question (by prompting an LLM and extracting logits for these tokens). These likelihoods allow us to perform a Bayesian update to obtain the posterior probability of $\omega$ given the answer, leading to probability distributions for the children of the current node.

**4. Explanation of Reward Design (Meaning of Accumulated Reward) (Proposed by reviewer1 Ho3e15, reviewer3 YbAg)**

Accumulated reward accounts for information gains over multiple rounds of the conversation. Suppose we are evaluating the reward for a 3-step conversation. The immediate reward function $R_u(v)$ accounts for the information gained from receiving the answer at node v. However, at the end of a 3-step conversation, the total information we have gained is actually the accumulation of information we have gained over the 3 conversation rounds. Thus, we accumulate the immediate reward $R_u(v)$ at a leaf node with those of its parent nodes, to compute the node $v$'s accumulated reward.

**We also provide more details about these major concerns and remaining questions in the below individual response.**

---

### Decision · Program_Chairs · 2024-09-25

**Decision:**

Accept (poster)

**Comment:**

Overall there is agreement that the paper is addressing an interesting problem and has offered a novel approach that yields strong empirical results.

There were a number of presentation and clarification issues raised by reviewers that the authors should work hard to include in the final paper.